# Fast T2T: Optimization Consistency Speeds Up Diffusion-Based Training-to-Testing Solving for Combinatorial Optimization

**Yang Li**[1†]**, Jinpei Guo**[1†]**, Runzhong Wang**[2]**, Hongyuan Zha**[3]**, Junchi Yan**[1] [*]
[1]Dept. of CSE & School of AI & MOE Key Lab of AI, Shanghai Jiao Tong University
[2]Massachusetts Institute of Technology
[3]The Chinese University of Hong Kong, Shenzhen
`{yanglily,mike0728,yanjunchi}@sjtu.edu.cn`
`runzhong@mit.edu, zhahy@cuhk.edu.cn`

## Abstract

Diffusion models have recently advanced Combinatorial Optimization (CO) as a powerful backbone for neural solvers. However, their iterative sampling process requiring denoising across multiple noise levels incurs substantial overhead. We propose to learn direct mappings from different noise levels to the optimal solution for a given instance, facilitating high-quality generation with minimal shots. This is achieved through an optimization consistency training protocol, which, for a given instance, minimizes the difference among samples originating from varying generative trajectories and time steps relative to the optimal solution. The proposed model enables fast single-step solution generation while retaining the option of multi-step sampling to trade for sampling quality, which offers a more effective and efficient alternative backbone for neural solvers. In addition, within the training-to-testing (T2T) framework, to bridge the gap between training on historical instances and solving new instances, we introduce a novel consistency-based gradient search scheme during the test stage, enabling more effective exploration of the solution space learned during training. It is achieved by updating the latent solution probabilities under objective gradient guidance during the alternation of noise injection and denoising steps. We refer to this model as Fast T2T. Extensive experiments on two popular tasks, the Traveling Salesman Problem (TSP) and Maximal Independent Set (MIS), demonstrate the superiority of Fast T2T regarding both solution quality and efficiency, even outperforming LKH given limited time budgets. Notably, Fast T2T with merely one-step generation and one-step gradient search can mostly outperform the SOTA diffusion-based counterparts that require hundreds of steps, while achieving tens of times speedup. The codes are publicly available at `https://github.com/Thinklab-SJTU/Fast-T2T`.

## 1 Introduction

Combinatorial Optimization (CO) problems, which involve optimizing discrete variables under given objectives, are essential in computer science and operational research. Due to the inherent computational difficulty, e.g. NP-hardness, solving efficiency poses significant challenges and requires exhaustive human efforts to design solving heuristics. Recent progress in this domain has shown promise in automatically learning heuristics with Machine Learning (ML) in a data-driven

[*]Correspondence author. † denotes equal contribution. This work was partly supported by NSFC (92370201, 62222607) and Shanghai Municipal Science and Technology Major Project under Grant 2021SHZDZX0102.

38th Conference on Neural Information Processing Systems (NeurIPS 2024).

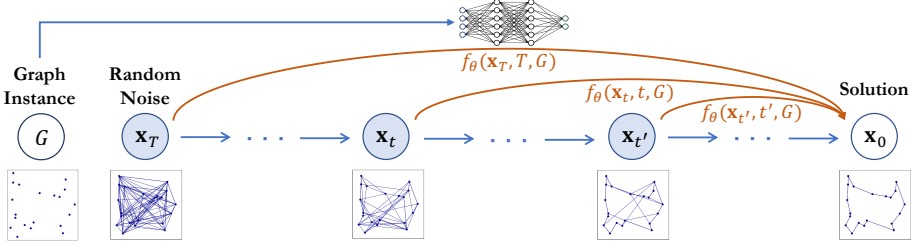

Figure 1: Optimization consistency models for CO solving where the model learns how to map from varying levels of noise to the solution distribution, conditioned on the problem graph instance.

manner [1, 2, 3, 4, 5, 6, 7, 8], bringing practical advantages in both quality and speed, especially when the instances are within a certain domain. In addition, learning can help quickly uncover new heuristics for new problems or new instance distributions where experts are not there.

Learning-based solvers for CO typically employ neural networks to generate neural predictions for solution construction or search guidance, aiming to minimize either the objective score [2, 4, 5, 6, 9] or the deviation from reference solutions [10, 3, 11, 12, 13]. The problem-solving task places significant demands on the testing performance of the model, while optimizing the average performance across training data does not ensure optimal performance for every encountered test instance. Thus, methods [14, 15, 6, 8] have been proposed to perform tailored optimization on neural predictions for every testing instance. In particular, generative modeling like diffusion has shown promise in learning instance-conditioned quality solution distributions [7, 8] with robust expressive power to achieve state-of-the-art performance, which also provides more informative support for further exploitation like gradient search in the solving stage, which was previously proposed as the diffusion-based training-to-testing (T2T) framework [8]. However, a major drawback of the diffusion backbone lies in its costly inference process, which necessitates tens or hundreds of denoising steps to solve one problem instance. This limitation in inference speed is crucial since CO seeks to achieve the highest solution quality within the shortest possible time, where both performance and efficiency are pivotal metrics in this pursuit. Although the diffusion solvers [7, 8] can exhibit superiority in inference speed compared to certain traditional methods and prior learning-based solvers, there remains substantial potential for speed enhancement, where bolstering this aspect could provide fundamental support and several-fold speedup for neural solvers based on generative modeling.

To resolve this issue, drawing inspiration from the successful practice of consistency models [16] for image generation, we propose the optimization consistency models to speed up the diffusion-based T2T framework, dubbed as Fast T2T, specifically for optimization problem-solving. We follow [8] to approach CO problems as conditional generation tasks, with the goal of modeling the distribution of high-quality solutions specific to given problem instances. As illustrated in Fig. 1, Fast T2T builds upon the methodology foundation of the discrete diffusion models [17, 18, 19] where a smooth transition from random uniform noise to the high-quality solution distribution is established. Given a problem instance, Fast T2T trains the conditional prediction consistency directly from varying noise levels to the solution distribution centered on the optimal solution to enable fast one-step solution distribution estimation. Meanwhile, to bridge the disparity between data-driven training and problem-solving, Fast T2T incorporates a novel objective gradient search for every instance in the testing phase based on the trained optimization consistency mappings.

Specifically, for the solving task, the model is expected to deliver the optimal solution output to the best extent possible for a given input instance. Thus, we define the optimization consistency property for the optimization scenario by conditional generation: *conditoned on a given instance $G$, points on all trajectories of all noising steps consistently map to the optimal solution of $G$.* Compared to the diffusion prediction of the data distribution from noising step $t$ to step $t-1$, the consistency modeling enables generating solutions ($\mathbf{x}_0$ in Fig. 1) from random noise vectors ($\mathbf{x}_T$ in Fig. 1) by a single step of model inference. This is achieved by an optimization consistency training protocol that minimizes the difference among samples originating from varying trajectories and noising steps relative to the optimal solution. The model retains the capability for multi-step sampling to trade for sampling quality by alternating noise introduction on $\mathbf{x}_0$ to generate a less noisy point $\mathbf{x}_t$ and solution reconstruction to obtain a new $\mathbf{x}_0$. Additionally, we design a novel objective gradient-based search

on top of the learned consistency mapping to further explore the learned solution distribution for every test instance. We introduce instance-specific guidance from the objective to the learned solution prior $p_\theta(\mathbf{x}|G)$ and obtain the posterior $p_\theta(\mathbf{x}|y^*, G)$ where $y^*$ represents the optimal objective score given instance $G$, thereby directing the sampling process to the optimal $\mathbf{x}^*$. It specifically entails minimizing the free energy corresponding to the posterior by updating the probability parameters of intermediate noisy points through exponential gradient updates guided by the objective function during the alternation of noise injection and denoising steps.

We show the efficacy of Fast T2T on two typical CO problems for edge-decision and node-decision types respectively, i.e., Traveling Salesman Problem (TSP) and Maximum Independent Set (MIS). We show that Fast T2T, even with a single-step initial solution generation and a single-step gradient search, can mostly outperform the SOTA diffusion-based counterparts with hundreds of inference steps. Meanwhile, due to its reduced step requirement, Fast T2T naturally demands significantly less inference time to achieve comparable quality, with more steps for further enhancement.

The highlights of this paper include: **1)** We introduce the optimization consistency condition and establish Fast T2T based on the proposed optimization consistency models to facilitate fast high-quality CO solving, which offers a highly effective and efficient backbone for learning-based solvers. **2)** To complement the learned prior and bridge the disparity between data-driven training and the requirement of problem-solving, we introduce a novel gradient search with objective guidance based on consistency mappings to conduct a tailored search for every test instance. **3)** Extensive experiments show that Fast T2T exhibits strong performance superiority over existing SOTA neural solvers on benchmark datasets across various scales.

## 2 Related Work

**Machine Learning for Combinatorial Optimization.** Current learning-based CO solvers can be categorized into constructive approaches and improvement-based approaches. Constructive approaches refer to autoregressive methods [20, 2, 4, 21, 5] that directly construct solutions by sequentially determining decision variables until a complete solution is constructed, and non-autoregressive methods [3, 12, 22, 6, 7, 23] that predict soft-constrained solutions in one shot and then perform post-processing to achieve feasibility. Improvement-based solvers [24, 25, 26, 27, 28] learn to iteratively refine a solution through local search operators toward minimizing the optimization objective.

Generative modeling for CO has recently shown promise with its potent representational capabilities and informative distribution estimation. It models the problem-solving task as a conditional generation task for learning solution distributions conditioned on given instances [21, 29, 7, 30, 31, 8]. Drawing from diffusion models, DIFUSCO [7] has attained SOTA performance in solving TSP and MIS. Nonetheless, it does not incorporate any instance-specific search paradigms to fully capitalize on the estimated solution distribution. Addressing this limitation, the T2T framework [8] further introduces an objective-guided gradient search process during solving to leverage the learned distribution. However, every aspect of this system, including distribution learning and gradient search, hinges on the diffusion model for step-by-step generation. This reliance renders the diffusion-based approaches computationally inefficient and impedes further search computations to trade for solution quality.

**Diffusion Models and Consistency Models.** Diffusion models entail a dual process comprising noise injection and learnable denoising, wherein neural networks predict the data distribution at each step based on the data from the previous step. For Diffusion in continuous space [17, 32, 33, 34, 35, 36, 37], the solution trajectories can be modeled by Probability Flow ODE [38]. Similar paradigms have also been adopted for discrete data using binomial or multinomial/categorial noises [17, 18, 19]. On top of the foundation of diffusion models, consistency models [16] define the self-consistency for every generation trajectory and introduce a consistency training paradigm for continuous data to directly learn the mappings from noise to the data. Inspired by this paradigm, we define the optimization consistency condition tailored for the optimization scenario, which requires consistency across multiple trajectories and time steps with the optimal solution as the target in a conditional context, thereby proposing the optimization consistency models as the solver embodiment. The models are employed on the discrete multinomial data for the benefit of CO.

# 3 Preliminaries and Problem Definition

Adopting the conventions established in [39, 40] we define $\mathcal{G}$ as the collection of CO problem instances represented by graphs $G(V, E) \in \mathcal{G}$, where $V$ and $E$ denote the nodes and edges respectively. CO problems can be broadly classified into two types based on the solution composition: edge-decision problems that involve determining the selection of edges and node-decision problems that determine nodes. Let $\mathbf{x} \in \{0, 1\}^{N \times 2}$ denote the optimization variable, where each entry is represented by a one-hot vector, i.e., each entry with $(0, 1)$ indicates that it is included in $\mathbf{x}$ and $(1, 0)$ indicates the opposite. For edge-decision problems, $N = n^2$ and $\mathbf{x}_{i,j}$ indicates whether $E_{i,j}$ is included in $\mathbf{x}$. For node-decision problems, $N = n$ and $\mathbf{x}_i$ indicates whether $V_i$ is included in $\mathbf{x}$. The feasible set $\Omega$ consists of $\mathbf{x}$ satisfying specific constraints as feasible solutions. A CO problem on $G$ aims to find a feasible $\mathbf{x}$ that minimize the given objective function $l(\cdot; G) : \{0, 1\}^{N \times 2} \to \mathbb{R}_{\geq 0}$:

$$\min_{\mathbf{x} \in \{0,1\}^{N \times 2}} l(\mathbf{x}; G) \quad \text{s.t.} \quad \mathbf{x} \in \Omega \tag{1}$$

TSP is defined on an undirected complete graph $G = (V, E)$, where $V$ represents $n$ cities and each edge $E_{i,j}$ is assigned a non-negative weight $w_{i,j}$ representing the distance between cities $i$ and $j$. The problem revolves around identifying a Hamiltonian cycle of minimum weight in $G$. For MIS, given an undirected graph $G = (V, E)$, an independent set is a subset of vertices $S \subseteq V$ such that no two vertices in $S$ are adjacent in $G$. MIS entails finding an independent set of maximum cardinality in $G$.

# 4 Training-Stage Optimization Consistency Modeling

## 4.1 Solution Encoding and Noising Process

Using the notations in Sec. 3, we represent the solutions of CO problems as $\mathbf{x} \in \{0, 1\}^{N \times 2}$ with $\mathbf{x} \in \Omega$. The distribution of $\mathbf{x}$ is represented by $N$ Bernoulli distributions indicating whether each entry should be selected, i.e., $p(\mathbf{x}) \in [0, 1]^{N \times 2}$. The objective of utilizing generative modeling for problem-solving is to capture the distribution of high-quality solutions conditioned on a given instance $G$, denoted as $p_\theta(\mathbf{x}|G)$. The neural models try to establish transition trajectories from random uniform noise to high-quality soft-constrained solutions, i.e., $\mathbf{x} \in \{0, 1\}^{N \times 2}$. These soft-constrained solutions are directly sampled from the estimated Bernoulli distributions where feasibility constraints can be broadly captured through learning and eventually hard-guaranteed by post-processing.

To establish the transition trajectories of data, we follow the discrete diffusion modeling [7, 8] to define the noising process, which takes the initial solution $\mathbf{x}_0$ sampled from the distribution $q(\mathbf{x}_0|G)$ and progressively introduces noise to generate a sequence of latent variables $\mathbf{x}_{1:T} = \mathbf{x}_1, \mathbf{x}_2, \cdots, \mathbf{x}_T$. Specifically, the noising process is formulated as $q(\mathbf{x}_{1:T}|\mathbf{x}_0) = \prod_{t=1}^{T} q(\mathbf{x}_t|\mathbf{x}_{t-1})$, which is achieved by multiplying $\mathbf{x}_t \in \{0, 1\}^{N \times 2}$ at step $t$ with a forward transition probability matrix $\mathbf{Q}_t \in [0, 1]^{2 \times 2}$ which indicates the transforming probability of decision state. We set $\mathbf{Q}_t = \begin{bmatrix} \beta_t & 1 - \beta_t \\ 1 - \beta_t & \beta_t \end{bmatrix}$ [18], where $\beta_t \in [0, 1]$ such that the transition matrix is doubly stochastic with strictly positive entries, ensuring that the stationary distribution is uniform which is an unbiased prior for sampling. The noising process for each step and the $t$-step marginal are formulated as:

$$q(\mathbf{x}_t|\mathbf{x}_{t-1}) = \text{Cat}(\mathbf{x}_t; \mathbf{p} = \mathbf{x}_{t-1}\mathbf{Q}_t) \quad \text{and} \quad q(\mathbf{x}_t|\mathbf{x}_0) = \text{Cat}(\mathbf{x}_t; \mathbf{p} = \mathbf{x}_0\overline{\mathbf{Q}}_t) \tag{2}$$

where $\text{Cat}(\mathbf{x}; \mathbf{p})$ is a categorical distribution over $N$ one-hot variables and $\overline{\mathbf{Q}}_t = \mathbf{Q}_1\mathbf{Q}_2 \cdots \mathbf{Q}_t$.

## 4.2 Optimization Consistency Training Scheme

Unlike the diffusion models modeling $p_\theta(\mathbf{x}_{t-1}|\mathbf{x}_t, G)$, we aim to directly map random noise to data by $p_\theta(\mathbf{x}_0|\mathbf{x}_t, G)$ in an optimization context. In continuous-time diffusion models defined on $(\epsilon, T)$ [38], consistency models [16] defines the self-consistency property as *points on the same trajectory map to the same initial point*, and optimize the learned consistency function $f_\theta(\cdot, \cdot)$ to satisfy the requirement by: 1) boundary condition: $f_\theta(\mathbf{x}_\epsilon, \epsilon) = \mathbf{x}_\epsilon$; 2) self-consistency property: $f_\theta$ outputs consistent estimation for arbitrary pairs of $(\mathbf{x}_t, t)$ that belong to the same trajectory, i.e., $f_\theta(\mathbf{x}_t, t) = f_\theta(\mathbf{x}_{t'}, t'), \forall\, t, t' \in [\epsilon, T]$. The joint effect of these two constraints serves as the necessary and sufficient condition to achieve a reliable data prediction from noise step $T$ to data, i.e., $f_\theta(\mathbf{x}_T, T) \to$

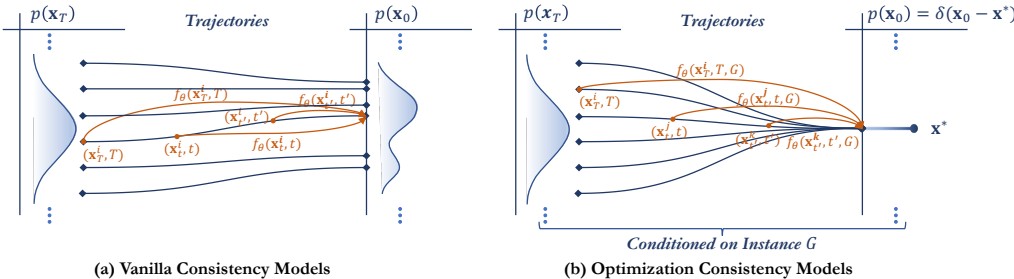

(a) Vanilla Consistency Models  (b) Optimization Consistency Models

Figure 2: Vannila consistency models are trained to map points on any trajectory to its origin. Optimization consistency enforces that all trajectories conditioned on $G$ consistently map to the same initial point, i.e., the optimal solution of $G$.

$\mathbf{x}_\epsilon$. In the optimization scenario of mapping instance $G$ to approximate its optimal solution $\mathbf{x}^*$, the generation process is conditioned on the problem instance $G$ with a reference optimal solution $\mathbf{x}^*$ serving as the commonly targeted initial point for all the conditional trajectories. Based on the discrete diffusion process with an explicit sampling process [18, 7, 8], we use the consistency function to estimate the optimal solution distribution as a point estimate $\delta(\mathbf{x} - \mathbf{x}^*)$ where $\delta(\cdot)$ represents Dirac delta. Below defines optimization consistency for the conditional context of problem-solving.

**Definition 4.1** (Optimization Consistency). Given a solution trajectory $\{\mathbf{x}_t\}_{t\in[0,T]}$, we define the consistency function as $f : (\mathbf{x}_t, t, G) \mapsto \delta(\mathbf{x} - \mathbf{x}^*)$, which maintains the optimization consistency property: conditioned on instance $G$, all points along any trajectory map to its optimal solution, i.e., $f_\theta(\mathbf{x}_t^i, t, G) = f_\theta(\mathbf{x}_{t'}^j, t', G) = \delta(\mathbf{x} - \mathbf{x}^*)$ for distinct trajectories $i$ and $j$ at distinct steps $t$ and $t'$.

As illustrated in Fig. 2, the goal of the consistency model $f_\theta$ in the optimization context, is to estimate the consistency function from data by learning to enforce optimization consistency. To achieve such consistency in the context of optimization to learn $f : G \mapsto \mathbf{x}^*$, given its nature as a conditional generation and the aim for an explicit optimal solution $\mathbf{x}^*$, we can seamlessly integrate $\mathbf{x}^*$ into the objective function for smooth training. Instead of optimizing the expectation of the variation of the consistency mappings over two noise points $\mathbf{x}$ and $\mathbf{x}'$, i.e., $\mathcal{L}_{\mathrm{CM}}(\theta) = \mathbb{E}\big[d\big(f_\theta(\mathbf{x}, t, G), f_\theta(\mathbf{x}', t', G)\big)\big]$, we introduce $\mathbf{x}^*$ to optimize the upper bound of $\mathcal{L}_{\mathrm{CM}}$ through triangle inequality of distance measures as

$$\mathcal{L}_{\mathrm{OptCM}}(\theta) = \mathbb{E}\big[d\big(f_\theta(\mathbf{x}, t, G), \delta(\mathbf{x} - \mathbf{x}^*)\big) + d\big(f_\theta(\mathbf{x}', t', G), \delta(\mathbf{x} - \mathbf{x}^*)\big)\big] \geq \mathcal{L}_{\mathrm{CM}}(\theta). \quad (3)$$

Here $d(\cdot, \cdot)$ is a distance metric function. In this case, the boundary conditions become less significant, since we have already dispersed the information of $\mathbf{x}^*$ across all noise time steps. Therefore, we can directly utilize the neural network $\theta$ to estimate the consistency function $f_\theta(\cdot, \cdot, \cdot)$. In addition, all learned trajectories are expected to map to the optimal solution $\mathbf{x}^*$ given the instance $G$, and the estimated solution distribution is expected to center on $\mathbf{x}^*$. This calls for the requirement of consistency extending across all trajectories, rather than being confined within a single trajectory.

**Definition 4.2.** The optimization consistency loss for conditional problem-solving is defined as:

$$\mathcal{L}_{\mathrm{OptCM}}^{N_t}(\theta) := \mathbb{E}\Big[\lambda(t_n)\Big(d\Big(f_\theta(\mathbf{x}_{t_n}^i, t_n, G), \delta(\mathbf{x} - \mathbf{x}^*)\Big) + d\Big(f_\theta(\mathbf{x}_{t_{n+1}}^j, t_{n+1}, G), \delta(\mathbf{x} - \mathbf{x}^*)\Big)\Big)\Big] \quad (4)$$

where the expectation is taken with respect to $G \sim p_G$, $n \sim \mathcal{U}[1, N_t - 1]$, $\mathbf{x}_{t_n}^i \sim \mathtt{Cat}(\mathbf{x}_{t_n}; \mathbf{p} = \mathbf{x}^*\overline{\mathbf{Q}}_{t_n})$, and $\mathbf{x}_{t_{n+1}}^j \sim \mathtt{Cat}(\mathbf{x}_{t_{n+1}}; \mathbf{p} = \mathbf{x}^*\overline{\mathbf{Q}}_{t_{n+1}})$. Here $\mathcal{U}[1, N_t - 1]$ denotes the uniform distribution over $\{1, 2, \cdots, N - 1\}$, $\lambda(\cdot) \in \mathbb{R}^+$ is a positive weighting function.

Since the model outputs $N$ Bernoulli distributions as the distribution of $\mathbf{x}_0$, we adopt the binary cross entropy to measure the distance between the estimation $p_\theta(\mathbf{x})$ and $\delta(\mathbf{x} - \mathbf{x}^*)$. We set $\lambda(t_n) \equiv 1$ and discover a decent empirical performance. $\mathbf{x}_{t_n}^i$ and $\mathbf{x}_{t_{n+1}}^j$ are identically and independently sampled from different noising trajectories, in comparison to $\mathbf{x}_{t_n}^i \sim \mathtt{Cat}(\mathbf{x}_{t_n}; \mathbf{p} = \mathbf{x}^*\overline{\mathbf{Q}}_{t_n})$, $\mathbf{x}_{t_{n+1}}^i \sim \mathtt{Cat}(\mathbf{x}_{t_{n+1}}; \mathbf{p} = \mathbf{x}_{t_n}\mathbf{Q}_{t_{n+1}} \cdots \mathbf{Q}_{t_{n+1}})$ where $\mathbf{x}_{t_n}^i$ and $\mathbf{x}_{t_{n+1}}^i$ are from the same trajectory. Since very close $t_n$ and $t_{n+1}$ would make Eq. 4.2 very easy to learn, we reschedule the time horizon into $N_t - 1$ sub-intervals $t_1 = 1 < t_2 < \cdots < t_{N_t} = T$ through the cosine denoising schedular such that $t_i = \lfloor \cos\big(\frac{1 - \pi \cdot ci}{2}\big) \cdot T \rceil$ following DDIM [34]. This training procedure enforces the model to learn

conditional consistency across different noise steps to consistently map to the optimal solution $\mathbf{x}^*$ of the given condition $G$. Note that although we enforce the noise to map to the Dirac delta on $\mathbf{x}^*$, the generative modeling process with a single sample per instance condition during training still enables the model to estimate a solution distribution (centering around the optimal solution) to enjoy diversity to enhance performance via parallel sampling, as evidenced by the experiments in Table. 2.

Specifically for implementation, the network $\theta$ is embodied as an anisotropic graph neural network with edge gating mechanisms [3], and instance $G$ serves as a part of the conditional input as the node or edge features. For TSP, the 2D coordinates of the vertices serve as the instance condition, and the input edge features are from the embeddings of entries in $\mathbf{x}_t$ integrated with the embedding of the input time step $t$. For MIS, the edges $E$ serve as the instance condition and the node embeddings are from $\mathbf{x}_t$ to collectively form the input. After the GNN iterations, the features of the decision variables (edges for TSP and nodes for MIS) are projected to 2-D outputs $p_\theta(\mathbf{x}_0|\mathbf{x}_t, G) \in [0,1]^{N \times 2}$ featuring $N$ Bernoulli distributions for $N$ entries in $\mathbf{x}_0$ via a linear layer followed by a Softmax layer.

## 5 Testing-Stage Problem Solving via Consistency-Based Gradient Search

The solving involves obtaining the initial solution from the raw consistency sampling process and a consistency-based gradient search process with objective feedback for iterative solution improvement.

### 5.1 Consistency Sampling for Initial Solutions

With a well-trained $f_\theta(\cdot, \cdot, \cdot)$, we generate solutions for a given instance $G$ by sampling $\mathbf{x}_T$ from the uniform distribution and then evaluate it for $\mathbf{x}_0 \sim p_\theta(\mathbf{x}_0) = f_\theta(\mathbf{x}_T, T, G)$. This process requires only one forward pass through the consistency model, resulting in sampling in a single step. Solution sampling with multiple steps of inferences can also be accomplished via alternating denoising and noise injection, allowing trading runtime for improved solving quality. Given a sequence of time points $\tau_1 > \tau_2 > \cdots > \tau_{N_\tau - 1}$, in time step $\tau_n$, the multistep sampling process adds noise to the $\mathbf{x}_0$ obtained from the last step $\tau_{n-1}$ by $\mathbf{x}_{\tau_n} \sim \mathtt{Cat}(\mathbf{x}_{\tau_n}; \mathbf{p} = \mathbf{x}\overline{\mathbf{Q}}_{\tau_n})$, then denoise to find the new solution by $\mathbf{x}_0 \sim f_\theta(\mathbf{x}_{\tau_n}, \tau_n, G)$, as shown in Algorithm. 1.

---

**Algorithm 1** Multistep Consistency Sampling

> **Input:** Consistency model $f_\theta(\cdot, \cdot, \cdot)$, graph problem instance $G$, sequence of time points $\tau_1 > \tau_2 > \cdots > \tau_{N_\tau - 1}$
> Sample $\mathbf{x}_T$ from uniform distribution $\mathcal{U}$
> $p_\theta(\mathbf{x}_0|G) \leftarrow f_\theta(\mathbf{x}_T, T, G)$
> $\mathbf{x}_0 \sim p_\theta(\mathbf{x}_0|G)$
> **for** $n = 1$ to $N_\tau - 1$ **do**
>     Sample $\mathbf{x}_{\tau_n} \sim \mathtt{Cat}(\mathbf{x}_{\tau_n}; \mathbf{p} = \mathbf{x}_0\overline{\mathbf{Q}}_{\tau_n})$
>     $p_\theta(\mathbf{x}_0|G) \leftarrow f_\theta(\mathbf{x}_{\tau_n}, \tau_n, G)$
>     $\mathbf{x}_0 \sim p_\theta(\mathbf{x}_0|G)$
> **end for**
> **Output:** Solution $\mathbf{x}_0$

---

### 5.2 Consistency-based Gradient Search with Objective Feedback

For CO, the integration of objective optimization facilitates direct engagement with the objective and enables efficient exploration of the solution space to minimize the score. [8] has established such a procedure for the step-by-step denoising function, yet it is not transferable to the consistency function, and incorporating objective optimization may prove more challenging as the consistency function maps across longer distance time steps. With the learned conditional solution prior $p_\theta(\mathbf{x}|G)$, this section aims to introduce a constraint $c(\mathbf{x}, y^*|G)$ on $\mathbf{x}$ to this prior for inference, where $y^*$ represents the optimal objective score given the instance $G$. That is, we want to find an approximation to the posterior distribution $p_\theta(\mathbf{x}|y^*, G) \propto p_\theta(\mathbf{x}|G)c(\mathbf{x}, y^*|G)$ to guide the sampling process to the optimal $\mathbf{x}^*$.

Here we follow [8] to determine $c(\mathbf{x}, y^*|G)$ by utilizing energy-based modeling [41] with the energy function $E(y, \mathbf{x}, G) = |y - l(\mathbf{x}; G)|$, which quantifies the compatibility between $y$ and $(\mathbf{x}, G)$, and it reaches zero when $y$ is exactly the objective score of $\mathbf{x}$ with respect to $G$. Such a design enables the best $y$ matching the inputs to maintain the highest probability density and the probability density is positively correlated with the matching degree. Then we employ the *Gibbs distribution* to characterize the probability distribution over a collection of arbitrary energies:

$$c(\mathbf{x}, y|G) = \frac{\exp(-E(y, \mathbf{x}, G))}{\int_{y'} \exp(-E(y', \mathbf{x}, G))} = Z\exp(-|y - l(\mathbf{x}; G)|) \tag{5}$$

Following [42], we introduce an approximate variational posterior $q(\mathbf{x}|G)$ and the free energy

$$F = \underbrace{-\mathbb{E}_{q(\mathbf{x}|G)q(\mathbf{h}|\mathbf{x},G)}\left[\log p_\theta(\mathbf{x},\mathbf{h}|G) - \log q(\mathbf{x})q(\mathbf{h}|\mathbf{x},G)\right]}_{F_1} \underbrace{-\mathbb{E}_{q(\mathbf{x}|G)}\left[\log c(\mathbf{x},y^*|G)\right]}_{F_2} \quad (6)$$

is minimized when $\mathrm{KL}(q(\mathbf{x}|G)\|p_\theta(\mathbf{x}|y^*,G))$ is minimized. Here $\mathbf{h} = \mathbf{x}_1,\cdots,\mathbf{x}_T$ represent the latent variables. Through the diffusion process, we can obtain $q(\mathbf{h}|\mathbf{x}) = \prod_{t=1}^{T} q(\mathbf{x}_t|\mathbf{x}_{t-1})$. We apply an approximation to the posterior over $\mathbf{x} = \mathbf{x}_0$ as a point estimate $q(\mathbf{x}|G) = \delta(\mathbf{x} - \boldsymbol{\eta})$. $F_1$ aligns with the objective of the consistency and diffusion models and $F_2$ can be transformed using Eq. 5:

$$F_1 = \mathbb{E}_{q(\mathbf{h}|\boldsymbol{\eta},G)}\left[\log \frac{q(\mathbf{h}|\boldsymbol{\eta},G)}{p_\theta(\boldsymbol{\eta},\mathbf{h}|G)}\right] \quad \text{and} \quad F_2 = -\log c(\boldsymbol{\eta},y^*|G) = l(\boldsymbol{\eta};G) - \log Z - y^*. \quad (7)$$

Initializing $\boldsymbol{\eta}$ from Sec. 5.1, we aim to update $\boldsymbol{\eta}$ to reach conditional solution distribution $p_\theta(\mathbf{x}|y^*,G)$ through exponential gradient decent on the latent continuous probability $\mathbf{p_x} = p(\mathbf{x}_{\alpha T}) = \boldsymbol{\eta}\overline{\mathbf{Q}}_{\alpha T} \in [0,1]^{N\times 2}$ at each iteration minimizing $F_1$ and $F_2$. Here $\mathbf{p_x}$ parameterizes $N$ Bernoulli distributions and $\alpha$ serves as a hyperparameter to control the noise degree. We view $\mathbf{p_x}$ as the expectation of $\mathbf{x}_{\alpha T}$ over $\mathbf{p_x}$, i.e., $\mathbb{E}_{\mathbf{p_x}}(\mathbf{x}_{\alpha T}) = \mathbf{p_x}$, since $\mathbf{p_x}$ is a multivariate Bernoulli. To obtain reliable gradients on $\mathbf{p_x}$, we estimate the expected distribution of $\mathbf{x}_0$ by $f_\theta(\mathbf{p_x}, \alpha T, G)$. Note $F_1$ is exactly the (implicit) objective of the diffusion and consistency models, i.e., the variational upper bound of the negative log-likelihood with the targeted data $\boldsymbol{\eta}$, which we optimize by minimizing the consistency over the re-predicted solutions $d\big(f_\theta(\mathbf{p_x}, \alpha T, G), \delta(\mathbf{x} - \boldsymbol{\eta})\big)$. While $F_2$ can be optimized by minimizing $l\big(f_\theta(\mathbf{p_x}, \alpha T, G); G\big) + \mathrm{Const}(\mathbf{p_x})$, where the objectives are defined following [8] as $l_{\mathrm{MIS}}(\mathbf{x}; G) \triangleq -\sum_{1\le i\le N} \mathbf{x}_i + \beta \sum_{(i,j)\in E} \mathbf{x}_i\mathbf{x}_j$ and $l_{\mathrm{TSP}} = \mathbf{x} \odot D$ where $D \in \mathbb{R}_+^{n\times n}$ denotes the distance matrix.

In each iteration, with current $\boldsymbol{\eta}$, we obtain $\mathbf{p_x} = \boldsymbol{\eta}\overline{\mathbf{Q}}_{\alpha T}$, $p_\theta(\boldsymbol{\eta}) = f_\theta(\mathbf{p_x}, \alpha T, G)$ and update $\mathbf{p_x}$ by

$$\mathbf{p_x} \leftarrow \mathbf{p_x} \odot \exp\big\{-\nabla_{\mathbf{p_x}}\big[\lambda_1 \cdot d\big(\mathbb{E}_{p_\theta(\boldsymbol{\eta})}\boldsymbol{\eta}, \delta(\mathbf{x} - \boldsymbol{\eta})\big) + \lambda_2 \cdot l\big(\mathbb{E}_{p_\theta(\boldsymbol{\eta})}\boldsymbol{\eta}; G\big)\big]\big\} \quad (8)$$

where $\lambda_1, \lambda_2$ are weighting hyperparameters. Then we sample $\mathbf{x}_{\alpha T} \sim \mathbf{p_x}$ and reconstruct a new distribution estimate of $\boldsymbol{\eta}$ by $p_\theta'(\boldsymbol{\eta}) = f_\theta(\mathbf{x}_{\alpha T}, \alpha T, G)$. To guarantee the feasibility, we utilize the logits of $p_\theta(\boldsymbol{\eta})$ and $p_\theta'(\boldsymbol{\eta})$ to produce the heatmaps where each element denotes each edge/node's confidence to be selected, and then adopt post-processing[2] to obtain two feasible solutions. This iteration concludes by outputting the lower-cost solution as $\boldsymbol{\eta}$.

## 6 Experiments

We test on two CO problems, TSP and MIS. The comparison includes SOTA learning-based solvers, heuristics, and exact solvers for each problem. To configure the generative-based models, we adopt $\mathrm{T_s}$ and $\mathrm{T_g}$ to represent the number of inference steps in initial solution sampling and the number of gradient search steps, respectively. For diffusion-based baselines including DIFUSCO [7] and T2T [8], we adopt $\mathrm{T_s}$=50 and involve 3 iterations with 5 guided denoising steps per iteration for T2T's gradient search, i.e., $\mathrm{T_g}$=15. Fast T2T can achieve promising results with merely one-step initial solution sampling and one-step gradient search, i.e., $\mathrm{T_s}$=1 and $\mathrm{T_g}$=1. However, the affordability of model inference facilitates a more extensive exploration of the solution distribution through a thorough search.

### 6.1 Experiments for TSP

**Datasets.** A TSP instance includes $N$ 2-D coordinates and a reference solution obtained by heuristics. Training and testing instances are generated via uniformly sampling $N$ nodes from the unit square $[0,1]^2$, which is a standard procedure as adopted in [2, 21, 3, 48, 6, 7, 8]. We experiment on various problem scales including TSP-50, 100, 500, and 1000.

**Metrics.** Following [2, 3, 6, 7, 8], we adopt three evaluation metrics: 1) Length: the average total distance or cost of the solved tours w.r.t. the corresponding instances, as directly corresponds to the objective. 2) Drop: the relative performance drop w.r.t. length compared to the global optimality or the reference solution; 3) Time: the average computational time to solve the problems.

---

[2]We follow previous works [6, 7, 8] to perform greedy decoding by sequentially inserting edges or nodes with the highest confidence if there are no conflicts. For TSP, the 2Opt heuristic [43] is optionally applied.

Table 1: Results with **Greedy Decoding** on TSP-50 and TSP-100. RL: Reinforcement Learning, SL: Supervised Learning, G: Greedy Decoding. * denotes results that are quoted from previous works.

| Algorithm | Type | TSP-50 | | | TSP-100 | | |
|---|---|---|---|---|---|---|---|
| | | Length↓ | Drop↓ | Time↓ | Length↓ | Drop↓ | Time↓ |
| Concorde [44] | Exact | 5.69 | 0.00% | (3m) | 7.76 | 0.00% | (12m) |
| LKH3 [45] | Heuristics | 5.69 | 0.00% | (3m) | 7.76 | 0.00% | (33m) |
| 2Opt [46] | Heuristics | 5.86 | 2.95% | – | 8.03 | 3.54% | – |
| AM* [2] | RL+G | 5.80 | 1.76% | (2s) | 8.12 | 4.53% | (6s) |
| GCN* [3] | SL+G | 5.87 | 3.10% | (55s) | 8.41 | 8.38% | (6m) |
| Transformer* [47] | RL+G | 5.71 | 0.31% | (14s) | 7.88 | 1.42% | (5s) |
| POMO* [4] | RL+G | 5.73 | 0.64% | (1s) | 7.84 | 1.07% | (2s) |
| Sym-NCO* [5] | RL+G | – | – | – | 7.84 | 0.94% | (2s) |
| Image Diffusion* [42] | SL+G | 5.76 | 1.23% | – | 7.92 | 2.11% | – |
| DIFUSCO ($T_s$=1) [7] | SL+G | 6.42 | 12.84% | (16s) | 9.32 | 20.20% | (20s) |
| DIFUSCO ($T_s$=50) [7] | SL+G | 5.71 | 0.45% | (9m) | 7.85 | 1.21% | (9m) |
| DIFUSCO ($T_s$=100) [7] | SL+G | 5.71 | 0.41% | (18m) | 7.84 | 1.16% | (18m) |
| Fast T2T ($T_s$=1) | SL+G | 5.71 | 0.31% | (11s) | 7.86 | 1.31% | (16s) |
| Fast T2T ($T_s$=3) | SL+G | 5.69 | 0.05% | (25s) | 7.77 | 0.17% | (33s) |
| Fast T2T ($T_s$=5) | SL+G | 5.69 | 0.02% | (1m) | 7.76 | 0.07% | (1m) |
| T2T ($T_s$=1,$T_g$=1) [8] | SL+G | 6.15 | 8.15% | (55s) | 9.00 | 16.09% | (1m) |
| T2T ($T_s$=50,$T_g$=15) [8] | SL+G | 5.69 | 0.07% | (18m) | 7.77 | 0.20% | (18m) |
| T2T ($T_s$=50,$T_g$=30) [8] | SL+G | 5.69 | 0.03% | (26m) | 7.76 | 0.11% | (42m) |
| Fast T2T ($T_s$=1,$T_g$=1) | SL+G | 5.69 | 0.03% | (54s) | 7.76 | 0.10% | (1m) |
| Fast T2T ($T_s$=2,$T_g$=2) | SL+G | 5.69 | 0.02% | (2m) | 7.76 | 0.04% | (2m) |
| Fast T2T ($T_s$=3,$T_g$=3) | SL+G | 5.69 | 0.01% | (3m) | 7.76 | 0.03% | (3m) |

**Results for TSP-50/100.** Given the recent success of learning-based solvers in achieving near-optimal performance on small-scale problems, we follow [8] to assess methods within the naive greedy decoding setting, aiming for a more discernable evaluation. The comparison includes state-of-the-art learning-based methods with greedy decoding and traditional solvers. Hyperparameter $\alpha$ is set as $0.2$. The sampling steps and gradient search steps are explicitly marked. Table. 1 shows that Fast T2T with merely one-step sampling steps approximates diffusion-based solvers with 100 sampling steps with a slight average performance gain of **5.7%**, yet with an average speedup of **82.8x**. A similar conclusion can be made for methods with gradient search with an average performance gain of **4.5%** and speedup of **35.4x**. Fast T2T variants with more sampling and gradient search steps achieve **82.1%** performance gain with **14.7x** speedup compared to previous state-of-the-art diffusion-based counterparts.

**Results for TSP-500/1000.** Learning-based solvers are compared using greedy decoding and sampling decoding ($\times$ 4), i.e., sampling multiple solutions and reporting the best one. Hyperparameter $\alpha$ is set as $0.2$. The sampling steps and gradient search steps are explicitly marked. Table. 2 shows that Fast T2T with merely one-step sampling steps averagely outperforms diffusion-based solvers with 100 sampling steps by a performance gain of **10.1%** and a speedup of **16.8x**. A similar conclusion can be made for methods with gradient search with an average performance gain of **14.9%** and a speedup of **8.5x**. Fast T2T variants with more sampling and gradient search steps achieve **52.1%** performance gain with **7.4x** speedup compared to previous SOTA diffusion-based counterparts.

**Results for Generalization.** Based on the problem set {TSP-50, TSP-100, TSP-500, TSP-1000}, we train the model on a specific problem scale and then evaluate it on all problem scales. Table 3 presents the generalization results of Fast T2T compared with diffusion-based counterparts with greedy decoding. The results show the satisfying cross-domain generalization ability of Fast T2T, e.g., the model trained on TSP-1000 achieves less than a 0.6% optimality gap on all other problem scales.

Table 3: Generalization results. *Tour length* and *drop* with **Greedy Decoding** are reported.

| Training \ Testing | | TSP-50 | TSP-100 | TSP-500 | TSP-1000 |
|---|---|---|---|---|---|
| TSP-50 | DIFUSCO ($T_s$=50)* [7] | 5.69, 0.09% | 5.70, 0.25% | 5.83, 2.55% | 5.84, 2.71% |
| | T2T ($T_s$=50,$T_g$=30)* [8] | 5.69, 0.02% | 5.70, 0.11% | 5.78, 1.60% | 5.75, 1.10% |
| | Fast T2T ($T_s$=5,$T_g$=5) | 5.69, 0.01% | 5.69, 0.02% | 5.71, 0.04% | 5.75, 1.02% |
| | Fast T2T ($T_s$=20,$T_g$=20) | 5.69, 0.00% | 5.69, 0.01% | 5.71, 0.21% | 5.73, 0.80% |
| TSP-100 | DIFUSCO ($T_s$=50)* [7] | 7.87, 1.44% | 7.78, 0.23% | 8.03, 3.44% | 8.02, 3.31% |
| | T2T ($T_s$=50,$T_g$=30)* [8] | 7.80, 0.55% | 7.77, 0.08% | 7.95, 2.47% | 7.91, 1.96% |
| | Fast T2T ($T_s$=5,$T_g$=5) | 7.77, 0.12% | 7.76, 0.02% | 7.79, 0.40% | 7.80, 0.55% |
| | Fast T2T ($T_s$=20,$T_g$=20) | 7.76, 0.08% | 7.76, 0.01% | 7.77, 0.23% | 7.78, 0.34% |
| TSP-500 | DIFUSCO ($T_s$=50)* [7] | 17.31, 4.61% | 17.05, 3.04% | 16.78, 1.40% | 16.86, 1.85% |
| | T2T ($T_s$=50,$T_g$=30)* [8] | 17.18, 3.79% | 16.92, 2.25% | 16.68, 0.81% | 16.72, 1.00% |
| | Fast T2T ($T_s$=5,$T_g$=5) | 16.99, 2.67% | 17.02, 2.87% | 16.61, 0.36% | 16.63, 0.51% |
| | Fast T2T ($T_s$=20,$T_g$=20) | 16.94, 2.34% | 16.97, 2.54% | 16.58, 0.20% | 16.60, 0.33% |
| TSP-1000 | DIFUSCO ($T_s$=50)* [7] | 24.17, 4.54% | 24.04, 3.98% | 23.65, 2.30% | 23.63, 2.21% |
| | T2T ($T_s$=50,$T_g$=30)* [8] | 24.20, 4.66% | 23.85, 3.16% | 23.47, 1.51% | 23.41, 1.23% |
| | Fast T2T ($T_s$=5,$T_g$=5) | 23.12, 3.43% | 24.08, 4.15% | 23.31, 0.82% | 23.25, 0.56% |
| | Fast T2T ($T_s$=20,$T_g$=20) | 23.86, 3.22% | 24.01, 3.87% | 23.25, 0.58% | 23.20, 0.36% |

**Soving Time vs. Optimality Drop on TSP-100/1000.** Fig. 3 and Fig. 4 illustrate the solving progress via the runtime-drop curves of Fast T2T and the prominent mathematical solver LKH3 [45]. The comparison is conducted on TSP-100 and TSP-1000. We are excited to discover that Fast T2T surpasses LKH3 in the early solving stage while also performing comparably in the later stage. This suggests that Fast T2T can serve as an effective rapid solver for approximate solutions outperforming

Table 2: Results on TSP-500 and TSP-1000. AS: Active Search, S: Sampling Decoding, BS: Beam Search. * denotes results that are quoted from previous works [8, 6].

| ALGORITHM | TYPE | TSP-500 | | | TSP-1000 | | |
|---|---|---|---|---|---|---|---|
| | | LENGTH↓ | DROP↓ | TIME | LENGTH↓ | DROP↓ | TIME |
| *Mathematical Solvers or Heuristics* | | | | | | | |
| Concorde [44] | Exact | 16.55 | 0.00% | 37.66h | 23.12 | 0.00% | 6.65h |
| Gurobi [49] | Exact | 16.55 | 0.00% | 45.63h | – | – | – |
| LKH-3 [45] | Heuristics | 16.55 | 0.00% | 46.28m | 23.12 | 0.00% | 2.57h |
| Farthest Insertion | Heuristics | 18.30 | 10.57% | 0s | 25.72 | 11.25% | 0s |
| *Learning-based Solvers with Greedy Decoding* | | | | | | | |
| AM* [2] | RL+G | 20.02 | 20.99% | 1.51m | 31.15 | 34.75% | 3.18m |
| GCN* [3] | SL+G | 29.72 | 79.61% | 6.67m | 48.62 | 110.29% | 28.52m |
| POMO+EAS-Emb* [15] | RL+AS+G | 19.24 | 16.25% | 12.80h | – | – | – |
| POMO+EAS-Tab* [15] | RL+AS+G | 24.54 | 48.22% | 11.61h | 49.56 | 114.36% | 63.45h |
| DIMES* [6] | RL+G | 18.93 | 14.38% | 0.97m | 26.58 | 14.97% | 2.08m |
| DIMES* [6] | RL+AS+G | 17.81 | 7.61% | 2.10h | 24.91 | 7.74% | 4.49h |
| DIMES* [6] | RL+G+2Opt | 17.65 | 6.62% | 1.01m | 24.83 | 7.38% | 2.29m |
| DIMES* [6] | RL+AS+G+2Opt | 17.31 | 4.57% | 2.10h | 24.33 | 5.22% | 4.49h |
| DIFUSCO ($T_s$=100) [7] | SL+G | 18.17 | 9.82% | 4m31s | 25.74 | 11.36% | 14m20s |
| Fast T2T ($T_s$=1) | SL+G | 17.80 | 7.57% | **17s** | 25.23 | 9.13% | **55s** |
| Fast T2T ($T_s$=5) | SL+G | 17.53 | 5.94% | 22s | 24.57 | 6.29% | 1m21s |
| T2T ($T_s$=50,$T_g$=30) [8] | SL+G | 17.48 | 5.61% | 6m23s | 25.21 | 9.04% | 19m21s |
| Fast T2T ($T_s$=1,$T_g$=1) | SL+G | 17.26 | 4.28% | 36s | 24.60 | 6.42% | **2m30s** |
| Fast T2T ($T_s$=5,$T_g$=5) | SL+G | **16.93** | **2.33%** | 2m12s | **23.96** | **3.64%** | 9m12s |
| DIFUSCO ($T_s$=100) [7] | SL+G+2Opt | 16.80 | 1.50% | 4m40s | 23.55 | 1.89% | 14m25s |
| Fast T2T ($T_s$=1) | SL+G+2Opt | 16.75 | 1.23% | **15s** | 23.45 | 1.42% | **57s** |
| Fast T2T ($T_s$=5) | SL+G+2Opt | **16.70** | **0.90%** | 22s | 23.38 | 1.14% | 1m20s |
| T2T ($T_s$=50,$T_g$=30) [8] | SL+G+2Opt | 16.68 | 0.82% | 6m29s | 23.44 | 1.40% | 19m39s |
| Fast T2T ($T_s$=1,$T_g$=1) | SL+G+2Opt | 16.67 | 0.73% | 39s | 23.35 | 1.00% | **2m33s** |
| Fast T2T ($T_s$=5,$T_g$=5) | SL+G+2Opt | **16.61** | **0.39%** | 2m10s | **23.25** | **0.58%** | 8m37s |
| *Learning-based Solvers with Sampling Decoding* | | | | | | | |
| EAN* [50] | RL+S+2Opt | 23.75 | 43.57% | 57.76m | 47.73 | 106.46% | 5.39h |
| AM* [2] | RL+BS | 19.53 | 18.03% | 21.99m | 29.90 | 29.23% | 1.64h |
| GCN* [3] | SL+BS | 30.37 | 83.55% | 38.02m | 51.26 | 121.73% | 51.67m |
| DIMES* [6] | RL+S | 18.84 | 13.84% | 1.06m | 26.36 | 14.01% | 2.38m |
| DIMES* [6] | RL+AS+S | 17.80 | 7.55% | 2.11h | 24.89 | 7.70% | 4.53h |
| DIMES* [6] | RL+S+2Opt | 17.64 | 6.56% | 1.10m | 24.81 | 7.29% | 2.86m |
| DIMES* [6] | RL+AS+S+2Opt | 17.29 | 4.48% | 2.11h | 24.32 | 5.17% | 4.53h |
| DIFUSCO ($T_s$=100) [7] | SL+S | 17.55 | 6.05% | 14m3s | 25.12 | 8.64% | 51m49s |
| Fast T2T ($T_s$=1) | SL+S | 17.63 | 6.56% | **53s** | 24.91 | 7.76% | **3m2s** |
| Fast T2T ($T_s$=5) | SL+S | 17.02 | 2.85% | 1m7s | 24.07 | 4.10% | 4m39s |
| T2T ($T_s$=50,$T_g$=30) [8] | SL+S | 17.04 | 2.99% | 19m33s | 24.85 | 7.49% | 49m42s |
| Fast T2T ($T_s$=1,$T_g$=1) | SL+S | 17.08 | 3.21% | 2m26s | 24.43 | 5.67% | **6m8s** |
| Fast T2T ($T_s$=5,$T_g$=5) | SL+S | **16.72** | **1.02%** | 7m9s | **23.68** | **2.44%** | 19m1s |
| DIFUSCO ($T_s$=100) [7] | SL+S+2Opt | 16.69 | 0.87% | 19m8s | 23.42 | 1.31% | 51m56s |
| Fast T2T ($T_s$=1) | SL+S+2Opt | 16.72 | 1.02% | **59s** | 23.39 | 1.17% | **3m12s** |
| Fast T2T ($T_s$=5) | SL+S+2Opt | **16.63** | **0.49%** | 1m8s | 23.30 | 0.77% | 4m50s |
| T2T ($T_s$=50,$T_g$=30) [8] | SL+S+2Opt | 16.63 | 0.48% | 19m42s | 23.37 | 1.07% | 51m3s |
| Fast T2T ($T_s$=1,$T_g$=1) | SL+S+2Opt | 16.64 | 0.54% | 2m33s | 23.31 | 0.83% | **6m14s** |
| Fast T2T ($T_s$=5,$T_g$=5) | SL+S+2Opt | **16.58** | **0.21%** | 6m51s | **23.22** | **0.42%** | 18m17s |

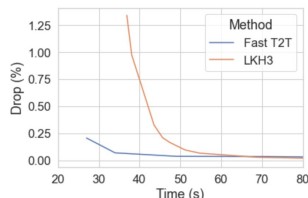

Figure 3: Effect of runtime to optimality drop for Fast T2T and LKH3 on TSP-100.

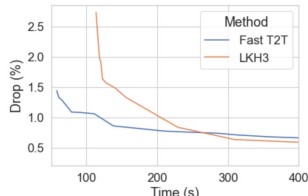

Figure 4: Effect of runtime to optimality drop for Fast T2T and LKH3 on TSP-1000.

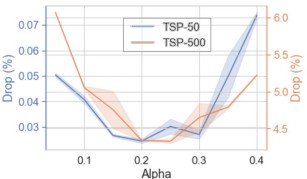

Figure 5: Effect of $\alpha$ to the performance drop.

LKH3, which may find widespread applications requiring prompt responses. Other neural solver baselines fall far outside the comparable range; please refer to Fig. 6 for an intuitive illustration.

**Ablation and Hyperparameter Study.** Fig. 5 illustrates the performance variation when altering the noise hyperparameter $\alpha$, and we discover a relatively superior and stable performance at $\alpha = 0.2$. Fig. 6 shows the performance variation when varying the sampling and gradient search steps. We also include DIFUSCO [7] and T2T [8] for direct comparison, in order to see whether diffusion-based methods can achieve promising results using minimal sampling steps. In this case, we let the gradient search steps equal to the sampling steps for Fast T2T and T2T. The results show a significant performance overwhelm of Fast T2T to diffusion-based counterparts.

## 6.2 Experiments for MIS

**Datasets.** Two datasets are tested for the MIS problem following [52, 54, 53, 6, 7], including RB graphs [31] and Erdős–Rényi (ER) graphs [55]. We randomly sample 200 to 300 vertices uniformly and generate the graph instances. ER graphs are randomly generated with each edge maintaining a fixed probability of being present or absent, independently of the other edges. We adopt ER graphs of 700 to 800 nodes with the pairwise connection probability set as 0.15.

**Metrics.** Following previous works [2, 3, 6, 7], we adopt three evaluation metrics to measure model performance: 1) Size: the average size of the solutions w.r.t. the corresponding instances, i.e. the objective. 2) Drop: the relative performance drop w.r.t. size compared to the optimal solution or the reference solution; 3) Time: the average computational time required to solve the problems.

**Main Results.** The baselines include SOTA neural methods with greedy and sampling decoding (×4), as well as exact solver Gurobi [49] and heuristic solver KaMIS [51]. The solving time of Gurobi is set as comparable to neural solvers, thus it does not reach optimality. Table. 4 shows that Fast T2T with merely one-step sampling and gradient search steps averagely approximates diffusion-based

Table 4: Results on MIS. TS: Tree Search, UL: Unsupervised Learning. * denotes results quoted from previous works [8, 31].

| ALGORITHM | TYPE | RB-[200-300] | | | ER-[700-800] | | |
|---|---|---|---|---|---|---|---|
| | | SIZE↑ | DROP↓ | TIME | SIZE↑ | DROP↓ | TIME |
| KaMIS [51] | Heuristics | 20.10* | – | 1h24m | 44.87* | – | 52.13m |
| Gurobi [49] | Exact | 19.98 | 0.01% | 47m34s | 41.28 | 7.78% | 50.00m |
| Intel [52] | SL+G | – | – | – | 34.86 | 22.31% | 6.06m |
| DIMES [6] | RL+G | – | – | – | 38.24 | 14.78% | 6.12m |
| DIFUSCO ($T_s$=100) [7] | SL+G | 18.52 | 7.81% | 16m3s | 37.03 | 18.53% | 5m30s |
| Fast T2T ($T_s$=1) | SL+G | 18.59 | 7.37% | **35s** | 36.72 | 18.17% | **11s** |
| Fast T2T ($T_s$=5) | SL+G | **18.74** | **6.65%** | 1m16s | **37.80** | **15.76%** | 24s |
| T2T ($T_s$=50,$T_g$=30) [8] | SL+G | 18.98 | 5.49% | 20m58s | 39.81 | 11.28% | 7m7s |
| Fast T2T ($T_s$=1,$T_g$=1) | SL+G | 19.37 | 3.51% | **1m18s** | 40.25 | 10.30% | **25s** |
| Fast T2T ($T_s$=5,$T_g$=5) | SL+G | **19.49** | **2.89%** | 4m44s | **40.68** | **9.34%** | 1m32s |
| Intel [52] | SL+TS | 18.47 | 8.11% | 13m4s | 38.80 | 13.43% | 20.00m |
| DGL [53] | SL+TS | 17.36 | 13.61% | 12m47s | 37.26 | 16.96% | 22.71m |
| LwD [54] | RL+S | – | – | – | 41.17 | 8.25% | 6.33m |
| GFlowNets [31] | UL+S | 19.18 | 4.57% | 32s | 41.14 | 8.53% | 2.92m |
| DIFUSCO ($T_s$=100) [7] | SL+S | 19.13 | 4.79% | 20m28s | 39.12 | 12.81% | 21m43s |
| Fast T2T ($T_s$=1) | SL+S | 18.91 | 5.81% | **42s** | 37.91 | 15.52% | **24s** |
| Fast T2T ($T_s$=5) | SL+S | **19.38** | **3.46%** | 1m50s | **39.81** | **11.27%** | 1m16s |
| T2T ($T_s$=50,$T_g$=30) [8] | SL+S | 19.38 | 3.53% | 30m18s | 41.41 | 7.72% | 27m45s |
| Fast T2T ($T_s$=1,$T_g$=1) | SL+S | 19.53 | 2.74% | **1m59s** | 40.98 | 8.66% | **1m19s** |
| Fast T2T ($T_s$=5,$T_g$=5) | SL+S | **19.70** | **1.90%** | 6m59s | **41.73** | **6.99%** | 5m51s |

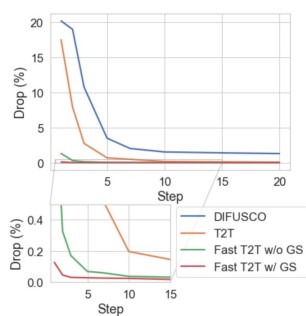

Figure 6: Effect of step number to drop for diffusion/consistency based methods. GS stands for gradient search.

counterparts with approximately 100 sampling steps by a slight performance gain of **2.5%** and a speedup of **26.3x**. Fast T2T variants with more sampling and gradient search steps achieve **23.7%** performance gain with **9.1x** speedup compared to previous SOTA diffusion-based counterparts.

## 7 Conclusion

We introduce optimization consistency on top of the diffusion-based training-to-testing solving framework for efficient and effective combinatorial optimization solving. Our proposed model facilitates rapid single-step solving, demonstrating comparable or superior performance to SOTA diffusion-based counterparts, offering a more effective and efficient alternative backbone for neural solvers. In addition, a novel consistency-based gradient search scheme is introduced to further complement the generalization capability during solving. Experimental results on TSP and MIS datasets showcase the superiority of our methods, exhibiting significant performance gains in both solution quality and speed compared to previous state-of-the-art neural solvers. Furthermore, our approach demonstrates superiority over LKH3 in the early stages of solving.

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

# Appendix

## A  Training Details

### A.1  Training Algorithm

---

**Algorithm 2** Optimization Consistency Training

---

1: **Input** dataset $\mathcal{D}$, consistency model $f_\theta(\cdot, \cdot)$, initial model parameter $\theta$, learning rate $\eta$, consistency loss function $d(\cdot, \cdot)$, inference steps $T$, scaling factor $\alpha$, Bernoulli noise matrix $\bar{Q}_{1,\dots,T}$, weighting function $\lambda(\cdot)$

2:

3: **repeat**

4:     Sample $\mathbf{x}^* \sim \mathcal{D}$, $t_1 \sim [1, T]$, $t_2 \leftarrow \lceil \alpha t_1 \rceil$

5:     Sample $\mathbf{z_1} \sim \mathbf{x}^* \bar{Q}_{t_1}$, $\mathbf{z_2} \sim \mathbf{x}^* \bar{Q}_{t_2}$

6:     $\mathcal{L}(\theta) \leftarrow \lambda(t_1)\left(d\left(\mathbf{f}_\theta\left(\mathbf{z_1}, t_1\right), \delta(\mathbf{z_1} - \mathbf{x}^*)\right)\right) + d\left(\mathbf{f}_\theta\left(\mathbf{z_2}, t_2\right), \delta(\mathbf{z_1} - \mathbf{x}^*)\right)$

7:     $\theta \leftarrow \theta - \eta \nabla_\theta \mathcal{L}(\theta)$

8:

9: **until** convergence

---

### A.2  Design Choices for Optimization Consistency

We supplement the specific design choices of the optimization consistency models, and the listed hyperparameters correspond to those used in the algorithm presented in sections 4 and 5.

| Training | Design Choice |
|---|---|
| Consistency Loss Function | $d(x, y) = \text{Binary\_Cross\_Entropy}(x, y)$ |
| Scaling Factor | $\alpha = 0.5$ |
| Weighting Function | $\lambda(t) = 1$ |
| Discretization Curriculum | $t \sim \{1, 2, \dots, T\}$, randomly sampling |
| Initial Learning Rate | $\eta = 0.0002$ |
| Learning Rate Schedule | Cosine decay, decay rate $\omega = 0.0001$ |

| Test | Design Choice |
|---|---|
| Sampling Step Schedule | $t_1 = T(1 - \sin(N \cdot i\pi/2))$, $t_2 = T(1 - \sin(N \cdot (i+1)\pi/2))$ |
| Guided Weighting Parameters | $\lambda_1 = 50, \lambda_2 = 50$ on TSP    $\lambda_1 = 2, \lambda_2 = 2$ on MIS |
| Rewrite Ratio | $\epsilon = 0.2$ on TSP and ER-[700-800]    $\epsilon = 0.3$ on RB-[200-300] |

## B  Supplementary Experiments

### B.1  Results on TSP Real-World Data

**Results on TSPLIB 50-200.** We evaluate our model trained with random 100-node problems on real-world TSPLIB instances with 50-200 nodes. The compared baselines include DIFUSCO [7], T2T [8], and baselines listed in [56]'s Table 3. The hyperparameter settings of the compared baselines are: DIFUSCO: $T_s$=50; T2T: $T_s$=50 and $T_g$=30; Fast T2T (w/o GS): $T_s$=10; Fast T2T (w/ GS): $T_s$=10 and $T_g$=10. The diffusion-based methods are compared in the same settings with greedy decoding and Two-Opt post-processing. For each instance, we normalize the coordinates to [0,1].

**Results on TSPLIB 50-200.** We also supplement the results (optimality drop) of diffusion-based baselines and Fast T2T on large-scale TSPLIB benchmark instances with 200-1000 nodes. The models are trained on TSP-500 and inference with greedy decoding and Two-Opt post-processing. For each instance, we normalize the coordinates to [0,1].

Table 5: Solution quality for methods trained on random 100-node problems and evaluated on **TSPLIB instances** with 50-200 nodes. * denotes results quoted from previous works [56].

| INSTANCES | AM* | GCN* | Learn2OPT* | GNNGLS* | DIFUSCO | T2T | Fast T2T (w/o GS) | Fast T2T (w/ GS) |
|---|---|---|---|---|---|---|---|---|
| eil51 | 16.767% | 40.025% | 1.725% | 1.529% | 2.82% | 0.14% | 0.00% | 0.00% |
| berlin52 | 4.169% | 33.225% | 0.449% | 0.142% | 0.00% | 0.00% | 0.00% | 0.00% |
| st70 | 1.737% | 24.785% | 0.040% | 0.764% | 0.00% | 0.00% | 0.01% | 0.00% |
| eil76 | 1.992% | 27.411% | 0.096% | 0.163% | 0.34% | 0.00% | 0.00% | 0.00% |
| pr76 | 0.816% | 27.793% | 1.228% | 0.039% | 1.12% | 0.40% | 0.00% | 0.00% |
| rat99 | 2.645% | 17.633% | 0.123% | 0.550% | 0.09% | 0.09% | 0.00% | 0.00% |
| kroA100 | 4.017% | 28.828% | 18.313% | 0.728% | 0.10% | 0.00% | 0.00% | 0.00% |
| kroB100 | 5.142% | 34.686% | 1.119% | 0.147% | 2.29% | 0.74% | 0.74% | 0.65% |
| kroC100 | 0.972% | 35.506% | 0.349% | 1.571% | 0.00% | 0.00% | 0.00% | 0.00% |
| kroD100 | 2.717% | 38.018% | 0.866% | 0.572% | 0.07% | 0.00% | 0.00% | 0.00% |
| kroE100 | 1.470% | 26.589% | 1.832% | 1.216% | 3.83% | 0.27% | 0.13% | 0.00% |
| rd100 | 3.407% | 50.432% | 1.725% | 0.003% | 0.08% | 0.00% | 0.00% | 0.00% |
| eil101 | 2.994% | 26.701% | 0.387% | 1.529% | 0.03% | 0.00% | 0.00% | 0.00% |
| lin105 | 1.739% | 34.902% | 1.867% | 0.606% | 0.00% | 0.00% | 0.00% | 0.00% |
| pr107 | 3.933% | 80.564% | 0.898% | 0.439% | 0.91% | 0.61% | 1.31% | 0.62% |
| pr124 | 3.677% | 70.146% | 10.322% | 0.755% | 1.02% | 0.60% | 0.08% | 0.08% |
| bier127 | 5.908% | 45.561% | 3.044% | 1.948% | 0.94% | 0.54% | 1.50% | 1.50% |
| ch130 | 3.182% | 39.090% | 0.709% | 3.519% | 0.29% | 0.06% | 0.00% | 0.00% |
| pr136 | 5.064% | 58.673% | 0.000% | 3.387% | 0.19% | 0.10% | 0.01% | 0.01% |
| pr144 | 7.641% | 55.837% | 1.526% | 3.581% | 0.80% | 0.50% | 0.39% | 0.39% |
| ch150 | 4.584% | 49.743% | 0.312% | 2.113% | 0.57% | 0.49% | 0.00% | 0.00% |
| kroA150 | 3.784% | 45.411% | 0.724% | 2.984% | 0.34% | 0.14% | 0.00% | 0.00% |
| kroB150 | 2.437% | 56.745% | 0.886% | 3.258% | 0.30% | 0.00% | 0.07% | 0.07% |
| pr152 | 7.494% | 33.925% | 0.029% | 3.119% | 1.69% | 0.83% | 1.17% | 0.19% |
| u159 | 7.551% | 38.338% | 0.054% | 1.020% | 0.82% | 0.00% | 0.00% | 0.00% |
| rat195 | 6.893% | 24.968% | 0.743% | 1.666% | 1.48% | 1.27% | 0.79% | 0.79% |
| d198 | 373.020% | 62.351% | 0.522% | 4.772% | 3.32% | 1.97% | 1.35% | 0.86% |
| kroA200 | 7.106% | 40.885% | 1.441% | 2.029% | 2.28% | 0.57% | 1.79% | 0.49% |
| kroB200 | 8.541% | 43.643% | 2.064% | 2.589% | 2.35% | 0.92% | 2.50% | 2.50% |
| **Mean** | 16.767% | 40.025% | 1.725% | 1.529% | 0.97% | 0.35% | 0.41% | 0.28% |

Table 6: Solution quality for methods trained on random 500-node problems and evaluated on **TSPLIB instances** with 200-1000 nodes.

| INSTANCES | DIFUSCO | T2T | Fast T2T (w/o GS) | Fast T2T (w/ GS) |
|---|---|---|---|---|
| a280 | 1.39% | 1.39% | 4.58% | 0.10% |
| d493 | 1.81% | 1.81% | 3.48% | 1.43% |
| d657 | 4.86% | 2.40% | 1.91% | 0.64% |
| fl417 | 3.30% | 3.30% | 7.45% | 2.01% |
| gil262 | 2.18% | 0.96% | 0.64% | 0.18% |
| lin318 | 2.95% | 1.73% | 2.24% | 1.21% |
| linhp318 | 2.17% | 1.11% | 2.00% | 0.78% |
| p654 | 7.49% | 1.19% | 4.84% | 1.67% |
| pcb442 | 2.59% | 1.70% | 1.47% | 0.61% |
| pr226 | 4.22% | 0.84% | 0.66% | 0.34% |
| pr264 | 0.92% | 0.92% | 0.77% | 0.73% |
| pr299 | 1.46% | 1.46% | 2.16% | 1.40% |
| pr439 | 2.73% | 1.63% | 0.53% | 0.50% |
| rat575 | 2.32% | 1.29% | 1.74% | 1.43% |
| rat783 | 3.04% | 1.88% | 1.76% | 1.03% |
| rd400 | 1.18% | 0.44% | 0.16% | 0.08% |
| ts225 | 4.95% | 2.24% | 3.31% | 1.37% |
| tsp225 | 3.25% | 1.69% | 0.84% | 0.81% |
| u574 | 2.50% | 1.85% | 1.31% | 0.94% |
| u724 | 2.05% | 2.05% | 2.15% | 1.41% |
| **Mean** | 2.87% | 1.59% | 2.20% | 0.93% |

## B.2 Results on MIS Real-World Data

We supplement the results on the SATLIB real-world dataset [57] below. Initially, we did not include the SATLIB results because Fast T2T requires more data to learn the consistency mapping, which, due to its greater power, is more challenging to learn. Unfortunately, SATLIB does not provide sufficient data for this purpose. However, we still discover a positive results of Fast T2T outperforming previous baselines.

Table 7: Results on MIS SATLIB dataset.

| Type | Method | Size | Drop | Time |
|------|--------|------|------|------|
| Heuristic | KAMIS | 425.96 | – | 37.58m |
| Gurobi | Exact | 425.95 | 0.00% | 26.00m |
| RL+Sampling | LwD | 422.22 | 0.88% | 18.83m |
| RL+Sampling | DIMES | 423.28 | 0.63% | 20.26m |
| UL+Sampling | GlowNets | 423.54 | 0.57% | 23.22m |
| SL+Sampling | DIFUSCO ($T_s = 100$) | 425.14 | 0.19% | 53m41s |
| SL+Sampling | T2T ($T_s = 50, T_g = 30$) | 425.18 | 0.18% | 38m1s |
| SL+Sampling | Fast T2T ($T_s = 5, T_g = 5$) | **425.23** | **0.17%** | 25m35s |

## B.3 Results for Generalization on TSP Datasets

Fig. 7 visualized the performance of DIFUSCO, T2T, and Fast T2T on different scales of TSP instances. The experimental settings are the same to Sec. 6.1.

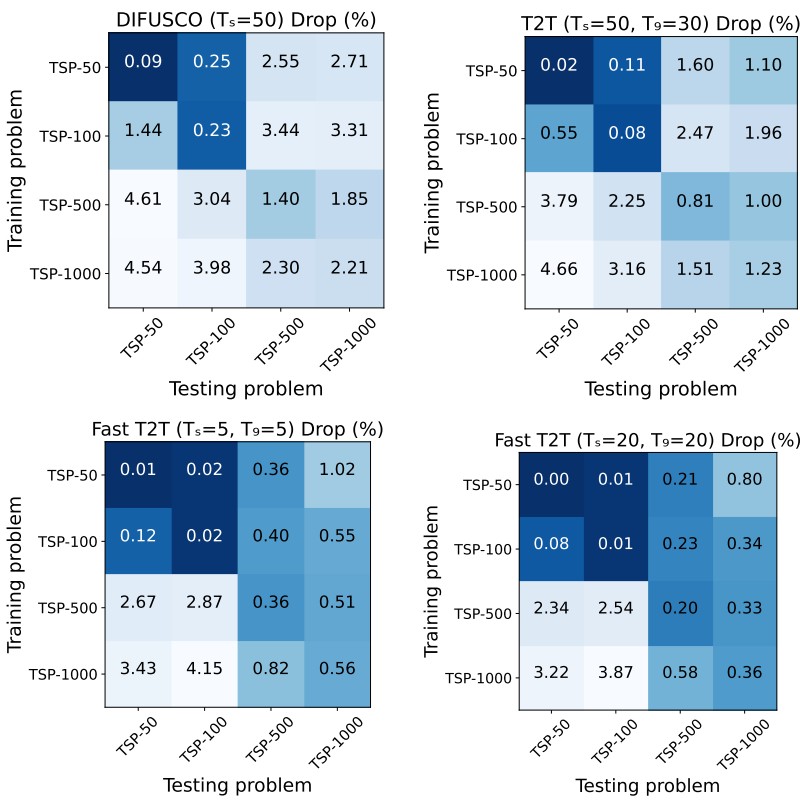

Figure 7: Confusion matrix of four scales from TSP datasets. Models are trained on scales on the $y$-axis, and tested with ***Greedy Decoding*** on scales on the $x$-axis. Values in matrices are the corresponding drop compared to exact solvers.

## B.4 Results for Generalization on MIS

We provide supplementary results for generalization results on the MIS problem below. We test the model trained on ER 700-800 with $p = 0.15$ to different $p$ (the probability that each simple edge exists) and $n$ (graph size). We find that the generalization ability of Fast T2T is significantly better than that of the previous diffusion-based methods DIFUSCO and T2T regarding both solution quality and speed, e.g., in ER 350-400 Sampling setting Fast T2T achieves significant performance gain from (23.28%, 24m31s) to (11.45%, 1m1s). Results are presented in Tables 8 and 9.

Table 8: Generalization Performance from $p = 0.15$ to $p = 0.2$, $p = 0.3$, and $p = 0.4$.

| p | Type | Method | Size | Drop | Time |
|---|---|---|---|---|---|
| 0.2 | Greedy | DIFUSCO ($T_s = 100$) | 26.25 | 25.65% | 6m31s |
| | | T2T ($T_s = 50, T_g = 30$) | 27.84 | 21.13% | 7m52s |
| | | Fast T2T ($T_s = 1, T_g = 1$) | 28.04 | 20.58% | 32s |
| | | Fast T2T ($T_s = 5, T_g = 5$) | 29.52 | 16.38% | 1m57s |
| | Sampling | DIFUSCO ($T_s = 100$) | 27.98 | 20.73% | 27m15s |
| | | T2T ($T_s = 50, T_g = 30$) | 28.07 | 20.49% | 33m58s |
| | | Fast T2T ($T_s = 1, T_g = 1$) | 28.81 | 18.39% | 1m40s |
| | | Fast T2T ($T_s = 5, T_g = 5$) | **30.10** | **14.74%** | 6m13s |
| 0.3 | Greedy | DIFUSCO ($T_s = 100$) | 15.84 | 34.99% | 7m58s |
| | | T2T ($T_s = 50, T_g = 30$) | 16.43 | 32.55% | 8m20s |
| | | Fast T2T ($T_s = 1, T_g = 1$) | 17.43 | 28.45% | 51s |
| | | Fast T2T ($T_s = 5, T_g = 5$) | 17.69 | 27.39% | 2m52s |
| | Sampling | DIFUSCO ($T_s = 100$) | 17.17 | 29.52% | 30m3s |
| | | T2T ($T_s = 50, T_g = 30$) | 16.38 | 32.78% | 37m27s |
| | | Fast T2T ($T_s = 1, T_g = 1$) | 17.79 | 26.97% | 2m2s |
| | | Fast T2T ($T_s = 5, T_g = 5$) | **18.38** | **24.53%** | 8m36s |
| 0.4 | Greedy | DIFUSCO ($T_s = 100$) | 11.75 | 35.40% | 9m40s |
| | | T2T ($T_s = 50, T_g = 30$) | 12.77 | 29.77% | 10m28s |
| | | Fast T2T ($T_s = 1, T_g = 1$) | 12.86 | 29.30% | 1m1s |
| | | Fast T2T ($T_s = 5, T_g = 5$) | 13.27 | 27.06% | 3m36s |
| | Sampling | DIFUSCO ($T_s = 100$) | 12.69 | 30.21% | 40m22s |
| | | T2T ($T_s = 50, T_g = 30$) | 13.03 | 28.33% | 45m2s |
| | | Fast T2T ($T_s = 1, T_g = 1$) | 13.31 | 26.80% | 2m12s |
| | | Fast T2T ($T_s = 5, T_g = 5$) | **13.56** | **25.43%** | 8m58s |

Table 9: Generalization Performance from ER 700-800 to ER 350-400 and 1400-1600.

| n | Decoding | Method | Size | Drop | Time |
|---|---|---|---|---|---|
| 350-400 | Greedy | DIFUSCO ($T_s = 100$) | 27.31 | 28.04% | 5m1s |
| | | T2T ($T_s = 50, T_g = 30$) | 28.54 | 24.80% | 6m59s |
| | | Fast T2T ($T_s = 1, T_g = 1$) | 32.56 | 14.20% | 22s |
| | Sampling | DIFUSCO ($T_s = 100$) | 29.33 | 22.73% | 20m12s |
| | | T2T ($T_s = 50, T_g = 30$) | 29.12 | 23.28% | 24m31s |
| | | Fast T2T ($T_s = 1, T_g = 1$) | 33.61 | 11.45% | 1m1s |
| 1400-1600 | Greedy | DIFUSCO ($T_s = 100$) | 34.39 | 32.48% | 22m7s |
| | | T2T ($T_s = 50, T_g = 30$) | OOM | OOM | OOM |
| | | Fast T2T ($T_s = 1, T_g = 1$) | 36.95 | 27.47% | 1m39s |
| | Sampling | DIFUSCO ($T_s = 100$) | 35.55 | 30.21% | 1h27m31s |
| | | T2T ($T_s = 50, T_g = 30$) | OOM | OOM | OOM |
| | | Fast T2T ($T_s = 1, T_g = 1$) | 38.59 | 24.25% | 3m56s |

We also supplement cross-dataset generalization results between RB graphs and ER graphs in Table 10. As seen, Fast T2T outperforms previous diffusion-based counterparts by a clear margin, e.g., in "Train:ER; Test:RB" "Sampling" setting, Fast T2T achieves significant performance gain from the previous (23.24%, 30m13s) to (9.10%, 4m20s).

## C Experimental Details

### C.1 Computational Resources.

Test evaluations on TSP-50/100 and MIS are performed on a single GPU of NVIDIA RTX 4090, and evaluations on TSP-500/1000 are performed on a single GPU of NVIDIA Telsla A100.

### C.2 Graph Sparsification.

For large-scale TSP problems, we follow [7, 8] to employ sparse graphs, as sparsified by constraining each node to connect to only its $k$ nearest neighbors, determined by Euclidean distances. For TSP-500,

Table 10: Performance Comparison Between Greedy and Sampling Methods (Train:ER; Test:RB).

| Setting | Type | Method | Size | Drop | Time |
|---|---|---|---|---|---|
| Train:ER; Test:RB | Greedy | DIFUSCO ($T_s = 100$) | 15.87 | 21.00% | 10m8s |
| | | T2T ($T_s = 50, T_g = 30$) | 16.59 | 17.41% | 15m5s |
| | | Fast T2T ($T_s = 1, T_g = 1$) | 16.73 | 16.59% | 40s |
| | | Fast T2T ($T_s = 5, T_g = 5$) | **17.01** | **15.21%** | 2m39s |
| | Sampling | DIFUSCO ($T_s = 100$) | 16.75 | 16.62% | 41m0s |
| | | T2T ($T_s = 50, T_g = 30$) | 16.80 | 16.40% | 29m48s |
| | | Fast T2T ($T_s = 1, T_g = 1$) | 17.29 | 13.78% | 57s |
| | | Fast T2T ($T_s = 5, T_g = 5$) | **17.38** | **13.38%** | 4m33s |
| Train:ER; Test:RB | Greedy | DIFUSCO ($T_s = 100$) | 29.98 | 27.54% | 10m48s |
| | | T2T ($T_s = 50, T_g = 30$) | 31.47 | 23.96% | 13m40s |
| | | Fast T2T ($T_s = 1, T_g = 1$) | 36.39 | 11.96% | 43s |
| | | Fast T2T ($T_s = 5, T_g = 5$) | **36.94** | **10.64%** | 2m37s |
| | Sampling | DIFUSCO ($T_s = 100$) | 31.67 | 23.47% | 44m0s |
| | | T2T ($T_s = 50, T_g = 30$) | 31.77 | 23.24% | 30m13s |
| | | Fast T2T ($T_s = 1, T_g = 1$) | 36.84 | 10.90% | 57s |
| | | Fast T2T ($T_s = 5, T_g = 5$) | **37.58** | **9.10%** | 4m20s |

we set $k = 50$, and for TSP-1000, $k = 100$. This strategy prevents the exponential increase in edges typical in dense graphs as node count rises.

## C.3 Datasets.

The reference solutions for TSP-50/100 are labeled by the Concorde exact solver [44] and the solutions for TSP-500/1000 are labeled by the LKH-3 heuristic solver [45]. The test set for TSP-50/100 is taken from [2, 3] with 1280 instances and the test set for TSP-500/1000 is from [12] with 128 instances for the fair comparison.

The reference solutions for both RB graphs and ER graphs are labeled with KaMIS [51]. For RB graphs, we randomly generate 90000 instances for the training set and 500 instances for the test set. For ER graphs, we randomly generate 163840 instaces for the training set and the test is from [6].

## C.4 Training Resource Requirement

We outline the offline training resource requirements of the Fast T2T framework in Table 11, with computations conducted on A100 GPUs. For contextual comparison, AM [2] necessitates 128M instances generated on-the-fly to train TSP-100, consuming 45.8 hours on 2 1080Ti GPUs. POMO [4] mandates 200M instances generated dynamically for TSP-100 training, entailing approximately one week on a single Titan RTX. Sym-NCO [5], an extension of POMO, requires approximately two weeks on a single A100 for training. Additionally, Sym-NCO [5] built upon AM [2] necessitates three days on 4 A100 GPUs. Compared with DIFUSCO [7], Fast T2T necessitates approximately double training time and GPU memory under the same settings, because our consistency training method requires forward twice for each training instance, leading to more time and memory consumption.

Table 11: Details about the training resource requirement of Fast T2T framework. The results are calculated on A100 GPUs.

| Problem Scale | Dataset Size | Batch Size | 1 GPU | 2 GPUs | 4 GPUs | GPU Mem |
|---|---|---|---|---|---|---|
| TSP-50 | 1,502 k | 32 | 112h 45m | 62h 24m | 41h 16m | 16.5 GB |
| TSP-100 | 1,502 k | 12 | 488h 12m | 268h 37m | 139h 26m | 23.2 GB |
| TSP-500 | 128 k | 6 | 142h 17m | 78h 58m | 45h 2m | 37.8 GB |
| TSP-1000 | 64 k | 4 | 324h 43m | 185h26s | 101h 3m | 20.2 GB |

## C.5 Hyperparamters

We conduct experiments on TSP and MIS benchmarks with our methods and compare the performance with prevalent learning-based solvers, heuristics, and exact solvers. The noise degree $\alpha$ associated with each benchmark is listed in Table. 12.

Table 12: Noise Degree for each benchmark.

| Benchmark | TSP-50 | TSP-100 | TSP-500 | TSP-1000 | RB-[200-300] | ER-[700-800] |
|---|---|---|---|---|---|---|
| $\alpha$ | 0.20 | 0.20 | 0.20 | 0.20 | 0.30 | 0.20 |

## C.6  Baseline Settings

### C.6.1  TSP Benchmarks

**TSP-50/100:** In the evaluation of TSP-50 and TSP-100, we compare our proposed Fast T2T against 11 baseline methods. These baselines include one exact solver, Concorde [44], two heuristic solvers - 2OPT [46] and Farthest Insertion - and seven learning-based solvers: AM [2], GCN [3], Transformer [47], POMO [4], Sym-NCO [5], Image Diffusion [42], DIFUSCO [7], and T2T [8]. Our post-processing involves greedy sampling and 2OPT refinement. To ensure equitable comparisons in terms of computational effort, we limit the number of inference steps for DIFUSCO to 100, and for T2T, we set the number of inference steps and guided search steps to 50 and 30, respectively.

**TSP-500/1000:** In the evaluation of TSP-500 and TSP-1000, our method is compared with 2 exact solvers, Concorde [44] and Gurobi [49], 2 heuristic solvers, LKH-3 [45] and Farthest Insertion, and 6 learning-based methods, including EAN [50], AM [2], GCN [3], POMO+EAS [15], DIMES [6], and DIFUSCO [7]. These learning-based methods can be further categorized into supervised learning (SL) and reinforcement learning (RL). Post-processing techniques employed encompass greedy sampling (Grdy, G), multiple sampling (S), 2OPT refinement (2OPT), beam search (BS), active search (AS), and combinations thereof. To ensure fair comparisons in terms of computational resources, we cap the number of inference steps for DIFUSCO at 100. Additionally, for T2T, we fix the number of inference steps and guided search steps at 50 and 30, respectively.

### C.6.2  MIS Benchmarks

We assess our method on two distinct benchmarks: RB-[200-300] and ER-[700-800]. Across both benchmarks, we compare the performance of Fast T2T against one exact solver, Gurobi [49], one heuristic solver, KaMIS [51], and 5 learning-based frameworks: Intel [52], DGL [52], LwD [54], DIMES [6], and DIFUSCO [7]. Post-processing strategies encompass greedy sampling (Grdy) and tree search (TS). Specifically, on both benchmarks, we set the number of inference steps at 100 for DIFUSCO. For T2T, we set the number of inference steps and guided search steps at 50 and 30, respectively.

# D  Network Architecture Details

## D.1  Input Embedding Layer

Given node vector $x \in \mathbb{R}^{N \times 2}$, weighted edge vector $e \in \mathbb{R}^E$, denoising timestep $t \in \{\tau_1, \ldots, \tau_M\}$, where $N$ denotes the number of nodes in the graph, and $E$ denotes the number of edges, we compute the sinusoidal features of each input element respectively:

$$\tilde{x}_i = \text{concat}(\tilde{x}_{i,0}, \tilde{x}_{i,1}) \tag{9}$$

$$\tilde{x}_{i,j} = \text{concat}\left( \sin \frac{x_{i,j}}{T^{\frac{0}{d}}}, \cos \frac{x_{i,j}}{T^{\frac{0}{d}}}, \sin \frac{x_{i,j}}{T^{\frac{2}{d}}}, \cos \frac{x_{i,j}}{T^{\frac{2}{d}}}, \ldots, \sin \frac{x_{i,j}}{T^{\frac{d}{d}}}, \cos \frac{x_{i,j}}{T^{\frac{d}{d}}} \right) \tag{10}$$

$$\tilde{e}_i = \text{concat}\left( \sin \frac{e_i}{T^{\frac{0}{d}}}, \cos \frac{e_i}{T^{\frac{0}{d}}}, \sin \frac{e_i}{T^{\frac{2}{d}}}, \cos \frac{e_i}{T^{\frac{2}{d}}}, \ldots, \sin \frac{e_i}{T^{\frac{d}{d}}}, \cos \frac{e_i}{T^{\frac{d}{d}}} \right) \tag{11}$$

$$\tilde{t} = \text{concat}\left( \sin \frac{t}{T^{\frac{0}{d}}}, \cos \frac{t}{T^{\frac{0}{d}}}, \sin \frac{t}{T^{\frac{2}{d}}}, \cos \frac{t}{T^{\frac{2}{d}}}, \ldots, \sin \frac{t}{T^{\frac{d}{d}}}, \cos \frac{t}{T^{\frac{d}{d}}} \right) \tag{12}$$

where $d$ is the embedding dimension, $T$ is a large number (usually selected as 10000), $\text{concat}(\cdot)$ denotes concatenation.

Next, we compute the input features of the graph convolution layer:

$$x_i^0 = W_1^0 \tilde{x}_i \tag{13}$$

$$e_i^0 = W_2^0 \tilde{e}_i \tag{14}$$

$$t^0 = W_4^0(\text{ReLU}(W_3^0 \tilde{t})) \tag{15}$$

where $t^0 \in \mathbb{R}^{d_t}$, $d_t$ is the time feature embedding dimension. Specifically, for TSP, the embedding input edge vector $e$ is a weighted adjacency matrix, which represents the distance between different nodes, and $e^0$ is computed as above. For MIS, we initialize $e^0$ to a zero matrix $0^{E \times d}$.

## D.2 Graph Convolution Layer

Following [3], the cross-layer convolution operation is formulated as:

$$x_i^{l+1} = x_i^l + \text{ReLU}(\text{BN}(W_1^l x_i^l + \sum_{j \sim i} \eta_{ij}^l \odot W_2^l x_j^l)) \tag{16}$$

$$e_{ij}^{l+1} = e_i^l + \text{ReLU}(\text{BN}(W_3^l e_{ij}^l + W_4^l x_i^l + W_5^l x_j^l)) \tag{17}$$

$$\eta_{ij}^l = \frac{\sigma(e_{ij}^l)}{\sum_{j' \sim i} \sigma(e_{ij'}^l) + \epsilon} \tag{18}$$

where $x_i^l$ and $e_{ij}^l$ denote the node feature vector and edge feature vector at layer $l$, $W_1, \cdots, W_5 \in \mathbb{R}^{h \times h}$ denote the model weights, $\eta_{ij}^l$ denotes the dense attention map. The convolution operation integrates the edge feature to accommodate the significance of edges in routing problems.

For TSP, we aggregate the timestep feature with the edge convolutional feature and reformulate the update for edge features as follows:

$$e_{ij}^{l+1} = e_{ij}^l + \text{ReLU}(\text{BN}(W_3^l e_{ij}^l + W_4^l x_i^l + W_5^l x_j^l)) + W_6^l(\text{ReLU}(t^0)) \tag{19}$$

For MIS, we aggregate the timestep feature with the node convolutional feature and reformulate the update for node features as follows:

$$x_i^{l+1} = x_i^l + \text{ReLU}(\text{BN}(W_1^l x_i^l + \sum_{j \sim i} \eta_{ij}^l \odot W_2^l x_j^l)) + W_6^l(\text{ReLU}(t^0)) \tag{20}$$

## D.3 Output Layer

The prediction of the edge heatmap in TSP and node heatmap in MIS is as follows:

$$e_{i,j} = \text{Softmax}(\text{norm}(\text{ReLU}(W_e e_{i,j}^L))) \tag{21}$$

$$x_i = \text{Softmax}(\text{norm}(\text{ReLU}(W_n x_i^L))) \tag{22}$$

where $L$ is the number of GCN layers and $\text{norm}$ is layer normalization.

## D.4 Hyper-parameters

For both TSP and MIS tasks, we construct a 12-layer GCN derived above. We set the node, edge, and timestep embedding dimension $d = 256, 128$ for TSP and MIS tasks, respectively.

# E Limitations and Broader Impacts

As the scale increases, our method's improvement in solving speed compared to diffusion-based methods will experience a certain degree of attenuation. This is because, with the expansion of the scale, the proportion of time required for relevant serial processing becomes larger, while the proportion of time for model inference is squeezed, resulting in a weakening of the speed improvement in the overall pipeline. This limitation can be addressed by combining our model with more efficient traditional solving strategies, which we leave for future work. Since the consistency model requires

two inference predictions with different noise levels during training, it requires twice the training cost of the original diffusion model. However, this overhead on training is offline, and the consistency model is much more efficient than diffusion at inference time.

Our work provides a more powerful and efficient backbone for neural combinatorial optimization, enabling significant performance improvement and versatility, making its application feasible across various solving frameworks. This work can be integrated into existing and future research in this field, driving progress in related studies.

