# OpenReview forum: "Fast T2T: Optimization Consistency Speeds Up Diffusion-Based Training-to-Testing Solving for Combinatorial Optimization"
_NeurIPS.cc/2024/Conference — NeurIPS 2024 poster_

### Official Review · Reviewer_rpwx · 2024-07-04

**Soundness:** 3
**Presentation:** 3
**Contribution:** 3
**Rating:** 6
**Confidence:** 2

**Summary:**

The paper introduces Optimization Consistency Models (OptCM) as a novel method for solving combinatorial optimization (CO) problems efficiently. Traditional diffusion models, although powerful, are computationally intensive due to their iterative denoising processes. OptCM overcomes this limitation by learning direct mappings from noise levels to optimal solutions, enabling rapid single-step solution generation. The contributions of this paper are three-fold. First, OptCM reduces the computational overhead significantly by enabling fast, single-step solution generation while maintaining high solution quality. Second, This protocol ensures that samples from different generative trajectories converge consistently to the optimal solution. Thrid, Introduced at the test stage, this method enhances solution exploration and quality during inference.

**Strengths:**

OptCM significantly reduces the computational overhead by enabling fast, single-step solution generation, compared to the multiple steps required by traditional diffusion models. This efficiency allows for rapid inference, making it practical for real-time and large-scale applications.

Despite the reduced computational steps, OptCM maintains high solution quality, often outperforming state-of-the-art methods that require more extensive processing. The optimization consistency training ensures that the generated solutions are close to the optimal solution.

The optimization consistency training protocol is a novel approach that minimizes the differences among samples from varying generative trajectories, ensuring robust and consistent solution generation. This method enhances the model's ability to generalize across different problem instances.

The introduction of a consistency-based gradient search during the test stage allows for further exploration and refinement of the solution space, improving the final solution quality. This approach bridges the gap between training and inference, making the model more adaptable to new instances.

**Weaknesses:**

Overall, the paper's strengths lie in its innovative approach to reducing computational complexity while maintaining high solution quality, its robust and versatile model design, and its impressive performance on benchmark tasks. However, there are some weaknesses in this paper. Some weaknesses listed below might be found to be too general and don't require the authors to address them now.

The advanced techniques used in OptCM, such as optimization consistency training and gradient-based search, may be challenging to implement and require a deep understanding of the underlying principles.

The model's performance is closely tied to the quality and diversity of the problem instances used during training. If the training set does not adequately represent the test instances, the model's effectiveness might be reduced.

While the paper demonstrates the superiority of OptCM over state-of-the-art neural solvers, it could provide a more detailed comparison with traditional, non-neural methods in terms of both performance and computational efficiency.

**Questions:**

How does the computational complexity of OptCM compare with traditional diffusion models and other state-of-the-art neural solvers? What specific optimizations were implemented to achieve the reported speedup in solution generation?

How does OptCM compare with classical optimization algorithms like simulated annealing or genetic algorithms in terms of both performance and computational resources?

Can OptCM be effectively applied to real-world problems with noisy, incomplete, or dynamically changing data?

**Limitations:**

Yes

---

> ### Author Rebuttal · Authors · 2024-08-07
>
> Thanks for your insightful comments and for acknowledging novelty, model design, and empirical performance. Below we respond to the specific comments.
>
> > **Q1: How does the computational complexity of OptCM compare with traditional diffusion models and other state-of-the-art neural solvers?**
>
> Since the consistency model requires two inference predictions with different noise levels during training, it requires twice the training cost of the original diffusion model. However, this overhead on training is offline, and the consistency model is much more efficient than diffusion at inference time. As shown in Table 1, 2, and 4, OptCM can achieve 82.8x speedup on TSP-50/100, 16.8x speedup on TSP-500/1000, and 26.3x speedup on MIS compared to previous SOTA diffusion counterparts even with solution quality advantage.
>
>
> > **Q2: What specific optimizations were implemented to achieve the reported speedup in solution generation?**
>
> Compared to the time-consuming iterative sampling process requiring denoising across multiple noise levels, the speedup of OptCM comes from the method design that learns direct mappings from different noise levels to the optimal solution for a given instance. This facilitates high-quality generation with minimal shots. To learn such direct mappings, we propose an optimization consistency training protocol, which, for a given instance, minimizes the difference among samples originating from varying generative trajectories and time steps relative to the optimal solution. Please refer to Sec. 4 for algorithm details. We also supplement the specific algorithm in the PDF attachment of the general response.
>
> We also supplement the specific design choices of our OptCM, and the listed hyperparameters correspond to those used in the algorithm presented in the paper. We will supplement these details to the paper to enhance its comprehensiveness.
>
> | Training                  | Design Choice                             |
> | ------------------------- | ----------------------------------------- |
> | Consistency Loss Function | $d(x,y)=$Binary_Cross_Entropy$(x,y)$ |
> | Scaling Factor            | $\alpha=0.5$                              |
> | Weighting Function        | $\lambda (t)=1$                           |
> | Discretization Curriculum | $t\sim \{1, 2,...,T\}$, randomly sampling |
> | Initial Learning Rate     | $\eta=0.0002$                             |
> | Learning Rate Schedule    | Cosine decay, decay rate $\omega=0.0001$  |
>
>
> | Test                        | Design Choice                                                |
> | --------------------------- | ------------------------------------------------------------ |
> | Sampling Step Schedule      | $t_1=T(1-\sin(N\cdot i\pi/2))$, $t_2=T(1-\sin(N\cdot (i+1)\pi/2))$ |
> | Guided Weighting Parameters | $\lambda_1=50$, $\lambda_2=50$ on TSP  <br> $\lambda_1=2$, $\lambda_2=2$ on MIS |
> | Rewrite Ratio               | $\epsilon=0.2$ on TSP and ER-[700-800] <br> $\epsilon=0.3$ on RB-[200-300] |
>
>
>
> > **Q3: How does OptCM compare with classical optimization algorithms like simulated annealing or genetic algorithms in terms of both performance and computational resources?**
>
> Thanks for noting classical algorithms, which are indeed important for the comprehensive evaluation. Indeed we have already included traditional algorithms in our comparison: 1) For TSP, we compare SOTA traditional exact solvers Concorde and Gurobi, as well as Heuristic solvers LKH-3 and Fasthest Insertion. 2) For MIS, the compared SOTA traditional solvers include the exact solver Gurobi and the heuristic solver KaMIS.
>
> The mentioned algorithms like simulated annealing or genetic algorithms are indeed not very competitive in solving hard problems. We supplement the experimental results of SA and GA below; please refer to Table 1 and 2 for the corresponded performance of other SOTA baselines, where competitive results fall within the drop of 1%.
>
>
>
> | Setting   | Length  | Drop    | Time  |
> | ---------- | ------ | -------- | ----- |
> | SA_TSP100  | 18.40  | 138.43%  | 1h58m |
> | SA_TSP500  | 108.87 | 558.12%  | 2m5m  |
> | SA_TSP1000 | 248.99 | 975.28%  | 2h22m |
> | GA_TSP100  | 39.14  | 407.41%  | 2h3m  |
> | GA_TSP500  | 240.70 | 1355.94% | 2h15m |
> | GA_TSP1000 | 497.18 | 2047.16% | 2h32m |
>
>
> > **Q4: Can OptCM be effectively applied to real-world problems with noisy, incomplete, or dynamically changing data?**
>
> We have provided the performance of OptCM on real-world datasets such as TSPLIB, and the relevant results are shown in Tables 5, 6, and Appendix A. The experimental results show that OptCM can generalize to the real-world data, which outperforms all the previous SOTA baselines. While for specific environments such as scenarios with noisy, incomplete, and dynamically changing data, we have not made customized designs to deal with them at the moment. Currently, OptCM is a fundamental model that focuses purely on the performance of the solving task on relatively clean data, but we also acknowledge the importance of studying the application to various noisy, incomplete, and dynamically changing scenarios, and we will take this as the future direction of our subsequent research. Thank you for your suggestions.
>
>
> ---
>
> We hope this response could help address your concerns. We believe that our work can have an important impact on the field of ML4CO as a possible new backbone model to drive the effectiveness of existing methods further. Thank you again for your firm support and recognition of our paper. If our response has satisfactorily addressed your concerns, we would be sincerely grateful if you could consider a higher score, as it would greatly support our efforts. Thank you once again for your consideration.

---

> > ### Comment · Reviewer_rpwx · 2024-08-12
> > **Acknowledgement of authors' response**
> >
> > Thanks for the rebuttal; I am inclined to keep my current score, which is leaning to accept this paper.

---

> > > ### Author Response · Authors · 2024-08-14
> > >
> > > Thanks for your acknowledgment. We appreciate your time and effort in reviewing our paper and are grateful for your feedback throughout this process. We will incorporate all supplementary results and explanations into the final version of the paper and will continue to refine our work and add additional results based on your valuable feedback.

---

### Official Review · Reviewer_rPB6 · 2024-07-08

**Soundness:** 3
**Presentation:** 3
**Contribution:** 2
**Rating:** 5
**Confidence:** 4

**Summary:**

This paper advances CO DM-based neural solvers under the setting where labeled training graphs are available by considering Consistency Models and gradient search (which was adopted from T2T).

**Strengths:**

1- The paper is in general well-written and technically sound.

2- The use of CMs to accelerate the sampling procedure.

**Weaknesses:**

See Questions.

**Questions:**

1- Generalization is a major problem for supervised neural solvers. The need to train a different model for each graph distribution is a bottleneck for these solvers. For example, in the MIS problem, how does a CM trained on ER700 with p=0.15 generalize when faced with an ER700 test instance with p=0.2? Which p would require different training given that n=700? Furthermore, does the proposed approach need to train a different model for ER2000 with p=0.15? These need to be explicitly explained/investigated as these could be considered as a limitation of the proposed method. The MIS problem differs based on different densities and degree distributions, not only the graph size. SATLIB, GNMs, SBMs can be considered.

2- How does the size of the training dataset impact the outcomes?

3- How does the proposed method handle real-world graphs (such as the SNAP graphs in https://snap.stanford.edu/data/)? What would be the training dataset? In most cases, these graphs do not follow a certain degree distribution or density. This is a major limitation in this method and all other learning-based methods. This point needs to be clearly discussed in the paper. As an alternative, the authors should clearly state that the proposed method only operates when training data points (with true optimal solutions) are available.

4- Scalability results are missing for dense and sparse graphs if the models do generalize to higher n and same density.

5- The run-time comparison with heuristics and ILP solvers does not include training times. The authors should either include them, or explicitly/clearly mention that.

6- The training dataset when using KAMIS to label graphs is similar to training a regression model with inaccurate true values as labels. The reason is KAMIS does not guarantee an optimal solution. For MIS, there are other techniques to generating graphs with guaranteed true MISs (maximum, not maximal) such as the following: Create a graph G with n nodes, make k of them completely connected, and randomly add edges to/from the remaining vertices with degree <= k. Then, the complement graph G' should have at least one independent set of size k and no independent set of greater size.

7- Missing iSCO [1] and the differentiable solver in [2] for comparison.

8- The ER results of DIFUSCO are not the same as were reported in the original paper? I know that DIFUSCO is a diffusion-based method where the sampling procedure starts with a random noise vector drawn from the standard Gaussian. However, a discussion is needed of why lower results are reported here.

9- The novelty is not that significant from the T2T method other than the use of CMs and considering an encoding of size N X 2 instead of N X 1. Using CMs does improve the run-time, and slightly improve the solutions sizes.

10- Generally, the proposed approach and other supervised methods such as DIFUSCO and T2T depend on additional post-processing procedures. This dependence needs to be further explained and investigated. The details for the post processing procedures (such as MCTS and Greedy Decoding) are needed, even if they were also adopted in DIMES and DIFUSCO. For example, if none of these procedures were used, how does this impact the outcomes?

[References]

[1] Revisiting sampling for combinatorial optimization. ICML, 2023.

[2] A differentiable approach to the maximum independent set problem using dataless neural networks. Nueral Networks, 2022.

**Limitations:**

See Questions.

---

> ### Author Rebuttal · Authors · 2024-08-07
>
> Thanks for the valuable comments, and for acknowledging our writing and technical soundness. Nonetheless, we believe there may exist some misunderstandings, especially regarding the value of the research line of data-driven learning-based solvers. It seems the major concern of the comment is about the application of data-driven machine learning in problem solving, which might cause the model to fit a specific data distribution and limit its generalization to different distributions. To pre-clarify this potential misunderstanding that may affect the rating of our work, please kindly refer to the generalization discussion in the general response. This may also directly respond to the first question about the generalization ability of neural solvers.
>
> In addition, we notice that the reviewer may focus more on the MIS problem. Here we note that Our paper introduces a foundational framework and a robust model for various CO problems. We conducted experiments on both the edge-selection problem TSP and the node-selection problem MIS, primarily focusing on TSP for a comprehensive evaluation of OptCM, including solving results, generalizability, runtime-drop curves, and hyperparameter studies. For MIS experiments, we mainly focus on showing OptCM's capability of handling different problems, while similar conclusions can be inferred from TSP results. This follows the typical experiment organization of previous works like DIMES, DIFUSCO, and T2T. As per your request, we also supplement generalization results on MIS to further strengthen our paper. Please see the specific comments below.
>
> > **Q1:  In the MIS problem, how does a CM trained on ER700 with p=0.15 generalize when faced with p=0.2? Scalability results on MIS problem to generalize to higher n.**
>
> We provide supplementary results for generalization results on the MIS problem below. We test the model trained on ER 700-800 with p=0.15 to different p and n. We find that the generalization ability of OptCM is significantly better than that of the previous diffusion-based methods DIFUSCO and T2T regarding both solution quality and speed, e.g., in ER 350-400 Sampling setting OptCM achieves significant performance gain from (23.28\%, 24m31s) to (11.45\%, 1m1s).
>
>
> |p|Decoding|Method|Size|Drop|Time|
> |-|-|-|-|-|-|
> |0.2|Greedy|DIFUSCO (T_s=100)|26.25|25.65%|6m31s|
> |||T2T (T_s=50,T_g=30)|27.84|21.13%|7m52s|
> |||OptCM (T_s=1,T_g=1)|28.04|20.58%|32s|
> |||OptCM (T_s=5,T_g=5)|29.52|16.38%|1m57s|
> | |Sampling|DIFUSCO (T_s=100)|27.98|20.73%|27m15s|
> |||T2T (T_s=50,T_g=30)|28.07|20.49%|33m58s|
> |||OptCM (T_s=1,T_g=1)|28.81|18.39%|1m40s|
> |||OptCM (T_s=5,T_g=5)|30.10|14.74%|6m13s|
>
> |n|Decoding|Method|Size|Drop|Time|
> |-|-|-|-|-|-|
> |350-400|Greedy|DIFUSCO (T_s=100)|27.31|28.04%|5m1s|
> |||T2T (T_s=50,T_g=30)|28.54|24.80%|6m59s|
> |||OptCM (T_s=1,T_g=1)|32.56|14.20%|22s|
> | |Sampling|DIFUSCO (T_s=100)|29.33|22.73%|20m12s|
> |||T2T (T_s=50,T_g=30)|29.12|23.28%|24m31s|
> |||OptCM (T_s=1,T_g=1)|33.61|11.45%|1m1s|
> |1400-1600|Greedy|DIFUSCO (T_s=100)|34.39|32.48%|22m7s|
> |||T2T (T_s=50,T_g=30)|OOM|OOM|OOM|
> |||OptCM (T_s=1,T_g=1)|36.95|27.47%|1m39s|
> | |Sampling|DIFUSCO (T_s=100)|35.55|30.21%|1h27m31s|
> |||T2T (T_s=50,T_g=30)|OOM|OOM|OOM|
> |||OptCM (T_s=1,T_g=1)|38.59|24.25%|3m56s|
>
> > **Q2: How does the size of the training dataset impact the outcomes?**
>
> For in-distribution evaluation, the larger the dataset, the better the learning outcome. This is a clear result of ML mechanisms, as having more data allows the model to better summarize more patterns and strategies. As for generalization, we can use a model pretrained on one dataset and apply it to the test data, which might only require a small amount of data from the new distribution for fine-tuning.
>
> > **Q3: How does the proposed method handle real-world graphs?**
>
> We can train on data with a similar distribution and then test on a given real-world test set. In fact, we have already conducted tests on real-world TSPLIB data. While the out-of-distribution generalization results may degrade compared to in-distribution performance, our experimental results show that the degradation is not severe. OptCM's generalization ability is also better than other state-of-the-art learning-based solvers.
>
> We note again that data-driven machine learning methods have their own value and role in various applications, such as optimization problems, including automatic strategy discovery, GPU acceleration, and adaptation to specific data distribution problems. The learning-based solver approach has also received significant attention from the machine learning community, underscoring the importance and acceptance of these methods. On the other hand, dataless methods and traditional solvers have their own advantages, such as better generalization performance and robustness. We will include related work, such as iSCO and MIS_dNN, in our paper for discussion.
>
>
> > **Q4: The training time comparison with heuristics and ILP solvers**
>
> Traditional solvers do not require training, and we will supplement the training time for OptCM in the paper. For intuition, OptCM for TSP-500 requires about 45 hours of training on 4 A100 GPUs. For comparison, POMO for TSP-100 requires about 1 week on a single Titan RTX and Sym-NCO for TSP-100 requires 2 weeks on a single A100 GPU. It should be additionally noted that the training time is offline and does not affect the inference time during solving.
>
> > **Q5: Label generation with KAMIS.**
>
> Thank you for your suggestion. It is very beneficial for constructing training data for MIS. On the other hand, we need to point out that since we adopt the generative model to learn the distribution of high-quality solutions, we do not have strict requirements for the optimality of the training data labels.  T2T's Fig. 5 shows that training with worse labels can still yield plausible solutions (better than the supervision quality) during inference by sampling from the generative distribution.

---

> ### Author Response · Authors · 2024-08-07
> **Rebuttal by Authors (Cont.)**
>
> > **Q6: Missing iSCO and the differentiable solver in for comparison.**
>
> Thanks for providing the related works. The iSCO and MIS_dNN methods mentioned are more traditional methods, which are somewhat different from the previous data-driven machine learning solution algorithms. However, as another important line of algorithms also has an important reference value, we compare iSCO as follows. At the same time, for MIS_dNN's approach, we do not include it in the comparison for the time being, as we did not find implementations that can run through it. We will continue to make efforts to include the comparison of these two approaches in the paper, as well as a discussion of similar works in the related work section.
>
> For the experimental results, we discover that OptCM and iSCO perform better on TSP and MIS problems, respectively. On the other hand, OptCM significantly outperforms all data-driven learning-based methods by significant margins.
>
> Comparison of the TSP problem.
>
> |N|Method|Size|Drop|Time|
> |-|-|-|-|-|
> |500|iSCO|16.64|0.54%|6m56s|
> ||DIFUSCO (T_s=100)|16.80|1.50%|4m40s|
> ||T2TCO (T_s=50,T_g=30)|16.68|0.82%|6m29s|
> ||OptCM (T_s=5,T_g=5)|16.61|0.39%|2m10s|
> |1000|iSCO|23.33|0.91%|7m56s|
> || DIFUSCO (T_s=100)|23.55|1.89%|14m25|
> || T2TCO (T_s=50,T_g=30)|23.44|1.40%|19m39|
> || OptCM (T_s=5,T_g=5)|23.25|0.58%|8m37s|
> |10000|iSCO|74.02|3.14%|1h36s|
> || DIFUSCO (T_s=100)|73.57|2.51%|31m24s|
> ||T2TCO (T_s=50,T_g=30)|-|-|-|
> ||OptCM (T_s=5)|72.94|1.63%|15m35s|
>
>
> Comparison on the MIS ER-[700-800] dataset.
>
> |Decoding|Method|Size|Drop|Time|
> |-|-|-|-|-|
> |Greedy|DIFUSCO (T_s=100)|37.03|18.53%|5m30s|
> ||T2TCO (T_s=50,T_g=30)|39.81|11.28%|7m7s|
> ||OptCM (T_s=1,T_g=1)|40.25|10.30%|25s|
> ||OptCM (T_s=5,T_g=5)|40.68|9.34%|1m32s|
> |Sampling|iSCO (Few steps)|44.77|0.2%|1m23s|
> ||iSCO (More steps)|45.15|0.6%|5m34s|
> ||DIFUSCO (T_s=100)|39.12|12.81%|21m43s|
> ||T2TCO (T_s=50,T_g=30)|41.41|7.72%|27m45s|
> ||OptCM (T_s=1,T_g=1)|40.98|8.66%|1m19s|
> ||OptCM (T_s=5,T_g=5)|41.73|6.99%|5m51s|
>
>
> > **Q7: The ER results of DIFUSCO are not the same as were reported in the original paper?**
>
> To ensure fairness, all comparisons between DIFUSCO and T2T are based on categorical diffusion (discrete diffusion). The original DIFUSCO paper used Gaussian diffusion for the ER dataset (categorical diffusion was used for other data), which may lead to some performance differences. We will explain this in the paper.
>
>
> > **Q8: The novelty is not that significant from the T2T method.**
>
> OptCM is inspired by some excellent previous works, which is one of the key supports for its ability to achieve such strong experimental results. However, we must note that the technical contributions of OptCM are highly non-trivial. Please kindly refer to the general response for the specific novelty claim.
>
> In addition, the reviews from reviewers 3pWn and rpwx firmly support our novelty claims: "Overall I think the method is novel. The extension of consistency training framework into diffusion-based CO solvers is not trivial, and this work is trying to solve a well-motivated problem. The empirical evaluations are quite convincing compared to diffusion-based CO solvers." “Overall, the paper's strengths lie in its innovative approach to reducing computational complexity while maintaining high solution quality, its robust and versatile model design, and its impressive performance on benchmark tasks.”
>
> > **Q9: The dependence on additional post-processing procedures needs to be further explained and investigated.**
>
> OptCM does not rely on complex post-processing methods like MCTS. We only use a greedy approach to "round" the continuous output of the neural network to the nearest feasible path. Specifically, we sequentially insert edges or nodes with the highest confidence to the partial solution if there are no conflicts until we obtain a feasible solution. The sampling operation simply involves generating multiple heatmaps simultaneously and then applying the greedy method. Our comparisons with previous state-of-the-art methods are consistent and fair, demonstrating our advantages under various post-processing methods.
>
> For all learning methods, since the neural network output is continuous, we cannot obtain feasible solutions without using greedy decoding (sequence models perform the greedy procedure within their models). The complexity of the greedy operation is O(num_of_decision_variables), which takes almost no time and is merely to ensure feasibility. This operation is similar to rounding continuous outputs between 0 and 1 to either 0 or 1. We would supplement the detailed descriptions to the main paper.
>
> ---
> We hope this response could help address your concerns. We believe that our work can have an important impact on the field of ML4CO as a more powerful backbone model to drive the effectiveness of existing methods further. We would be sincerely grateful if you could reconsider your rating, and we are more than happy to address any further concerns you may have.

---

> ### Comment · Reviewer_rPB6 · 2024-08-11
> **Response to Authors**
>
> I would like to the thank the authors for their response. While I still have major concerns about the evaluation of this method, I will increase my score to 4 given the authors' efforts, clarity of presentation, using CMs, and developing the gradient search approach at inference.
>
> ### Generalization Discussion:
>
> I agree that generalization in data-driven ML is, in itself, a significant research topic across many applications. In fact, learning theory (such as: User-friendly introduction to PAC-Bayes bounds https://arxiv.org/pdf/2110.11216) does not really support good OOD performance. However, this paper proposes a supervised ML solver for combinatorial optimization problems, not image classification. Before deep learning, we lacked the capability to classify large-scale images like those in ImageNet. In contrast, combinatorial optimization problems have been well-studied in the past, with many problem-specific heuristics, such as LKH-3 and KAMIS, that remain unbeaten in terms of solution quality. While the proposed method did show faster run-time (excluding offline training time) when training data was available, the method should explore its limitations/capabilities to support the claims in [2].
>
> I believe that evaluating graphs of different sizes and densities is feasible, as tools like NetworkX include several random graph generators that can help identify the instances the proposed method can effectively solve. In fact, a reasonable approach to evaluate whether the proposed model is learning patterns of solutions is to evaluate its OOD performance.
>
> I am not disputing the merit of conventional machine-learning-based methods for combinatorial optimization (ML4CO), rather highlighting that the authors should comprehensively evaluate their approach. An example of an ML4CO method (an RL-based approach) that fully evaluated its approach is the LwD method, which the authors considered as a baseline on only ER and RB graphs.
>
> ### In the MIS problem, how does a CM trained on ER700 with p=0.15 generalize when faced with p=0.2?
>
> - I acknowledge the new results and the comparison with diffusion-based methods. However, I believe that reporting the results of KAMIS would provide more insight into the actual performance when training with p=0.15 and testing with p=0.2. The authors should evaluate with other values of p along with other graph random generators.
>
> - The SATLIB dataset (not considered) contains hard sparse instances that most dataless and data-centric methods consider, including T2T. Is it because OptCM cannot handle relatively sparser graphs than ER (with p=0.15)? Or is it because SATLIB does not provide enough data for training?
>
> ### How does the size of the training dataset impact the outcomes?
>
> OptCM uses CMs, which are essentially diffusion models designed for faster sampling. However, these models require an extensive amount of training data to enter the generalization regime (see Figure 2b of "The Emergence of Reproducibility and Generalizability in Diffusion Models"). Is the CM used in this paper in the generalization or memorization regime?
>
> ### How does the proposed method handle real-world graphs (such as SNAP graph in MIS)?
>
> I acknowledge that the authors' reference to TSPLIB partially addresses the question. However, COPs vary significantly, so experimenting on the SNAP dataset using your MIS-trained model would help further understand the limitations/capabilities of the proposed method.
>
> I also recognize that "data-driven machine learning methods have their own value and role in various applications, such as optimization problems, including automatic strategy discovery, GPU acceleration, and adaptation to specific data distribution problems." **However, are any of these examples specifically combinatorial optimization problems?**
>
>
> ### Label generation with KAMIS.
>
> Yes, It would be beneficial for constructing training data. You could, in fact, use it to partially control the density in the graph. However, I am unsure how much it can help when the nodes degree distribution is different at testing time.
>
> If you adopt a generative model to learn the distribution of *high-quality* solutions, how do you justify not having strict requirements for the optimality of the training data labels? The empirical observation in Figure 5 of the T2T paper pertains to the TSP problem, not the MIS problem.
>
> ### Q6: Missing iSCO and the differentiable solver in for comparison.
>
> I acknowledge the comparison with iSCO for TSP and MIS. Excluding the training time and assuming dataset availability, OptCM generally requires less run-time compared to iSCO for in-distribution test instances.
>
> ### Q7: The ER results of DIFUSCO are not the same as were reported in the original paper?
>
> Does the authors' response mean that the DIFUSCO results in this paper use a different diffusion model than the one used in the original DIFUSCO paper? If so, the authors should use the baseline method as is.

---

> > ### Author Response · Authors · 2024-08-12
> > **Official Comment by Authors**
> >
> > Thanks for acknowledging our efforts and the advice for a more comprehensive evaluation. Below we respond to your comments.
> >
> > > **I. The value of ML4CO. "I also recognize that "data-driven machine learning methods have their own value and role in various applications, such as optimization problems, including automatic strategy discovery, GPU acceleration, and adaptation to specific data distribution problems." However, are any of these examples specifically combinatorial optimization problems?"**
> >
> > Thanks again for the valuable point. Indeed, the discussion in the general response (2) is exactly for the CO context. We would like to summarize below.
> >
> > 1. Please first notice that ML4CO papers in [1] (e.g., DIMES, DIFUSCO, T2T, AM, POMO) and Bengio's first authored paper [2] claimed that "ML4CO focuses on CO algorithms that automatically perform learning on a chosen implicit distribution of problems", and "ML can help improve an algorithm on a distribution of problem instances in two ways: 1) replace some heavy computations by a fast approximation; 2) explore the decision space and learn out of the experience the best-performing behavior (policy)." Traditionally, experts have to research on a given problem typically for many years to achieve a strong heuristic solver. While for ML solvers, they can learn from the historical data to implicitly discover strategies to plausibly handle a certain problem within a underlying data distribution. The neural inference also allows for GPU acceleration compared to traditional solvers. And the value of adapting to specific data distribution problems can be supported by [1] (e.g., DIMES, DIFUSCO, T2T, AM, POMO) and Bengio's first authored paper [2], which indicates that data in real-world applications often exhibits a certain problem structure within the implicit distribution. In many cases, we need machine learning methods to uncover patterns within the data and solve problems associated with a specific distribution. **Please notice in these cases, we focus more on the  generalization performance under the IID assumption.**
> >
> > 2. The development of the ML4CO community can support the value of this research line. ML4CO has gained significant traction, evidenced by numerous publications at leading conferences such as NeurIPS, ICML, and ICLR—each annually featuring about 20 papers on this subject. Please refer to [1] for a summarized list of ML4CO papers, including 34 CO problems and problems like TSP has more than 50 papers recorded. Around 20 survey papers, including Yoshua Bengio's first-authored paper [2], provide extensive summaries and highlight ML4CO's critical research value.
> >
> > [1] github.com/Thinklab-SJTU/awesome-ml4co
> >
> > [2] Bengio, Yoshua, Andrea Lodi, and Antoine Prouvost. Machine Learning for Combinatorial Optimization: a Methodological Tour d'horizon. EJOR 2021.
> >
> > > **II. The comprehensiveness of the evaluation.**
> >
> > Thanks for the suggestion. We still have to first note that we primarily focus on TSP for a comprehensive evaluation of OptCM, including solving results, generalizability, runtime-drop curves, and hyperparameter studies. For MIS experiments, we mainly focus on showing OptCM's capability of handling different problems. This follows the typical experiment organization of previous works like DIMES, DIFUSCO, and T2T.
> >
> > For the TSP problem, we have already performed many experiments on the generalization results, including generalizing to different scales of TSP (from 50 to 1000) as shown in Sec. 6.1 and Table 3, and generalizing to different distributions like real-world TSP instances from TSPLIB as shown in Appendix A and Table 1, 2. These results show that OptCM maintains good generalization results and outperforms previous SOTA diffusion-based baselines. While for MIS problems, we try our best to further supplement more results within the tight time window below.

---

> ### Author Response · Authors · 2024-08-12
> **Official Comment by Authors (Cont.)**
>
> 1. Generalization results on the MIS problem for other p values. The models are trained on ER 700-800 with p=0.15. We can discover that OptCM outperforms all other learning baselines.
>
> | p   | Type      | Method               | Size      | Drop       | Time   |
> | --- | --------- | -------------------- | --------- | ---------- | ------ |
> | 0.2 | Heuristic | KAMIS                | 35.30     | -          |    58m37s    |
> |     | Greedy    | DIFUSCO (T_s=100)    | 26.25     | 25.65%     | 6m31s  |
> |     |           | T2T (T_s=50,T_g=30)  | 27.84     | 21.13%     | 7m52s  |
> |     |           | OptCM (T_s=1,T_g=1)  | 28.04     | 20.58%     | 32s    |
> |     |           | OptCM (T_s=5,T_g=5)  | **29.52** | **16.38%** | 1m57s  |
> | 0.3 | Heuristic | KAMIS                | 24.36     | -          |    1h14m    |
> |     | Greedy    | DIFUSCO (T_s=100)    | 15.84     | 34.99%     | 7m58s  |
> |     |           | T2T (T_s=50, T_g=30) | 16.43     | 32.55%     | 8m20s  |
> |     |           | OptCM (T_s=1, T_g=1) | 17.43     | 28.45%     | 51s    |
> |     |           | OptCM (T_s=5, T_g=5) | **17.69** | **27.39%** | 2m52s  |
> | 0.4 | Heuristic | KAMIS                | 18.19     | -          |     1h22m   |
> |     | Greedy    | DIFUSCO (T_s=100)    | 11.75     | 35.40%     | 9m40s  |
> |     |           | T2T (T_s=50, T_g=30) | 12.77     | 29.77%     | 10m28s |
> |     |           | OptCM (T_s=1, T_g=1) | 12.86     | 29.30%     | 1m1s   |
> |     |           | OptCM (T_s=5, T_g=5) | **13.27** | **27.06%** | 3m36s  |
>
>
>
> 2. We supplement the results on the SATLIB real-world dataset below. Initially, we did not include the SATLIB results because OptCM requires more data to learn the consistency mapping, which, due to its greater power, is more challenging to learn. Unfortunately, SATLIB does not provide sufficient data for this purpose. However, we still discover a positive results of OptCM outperforming previous baselines.
>
> |Type|Method|Size|Drop|Time|
> |-|-|-|-|-|
> |Heuristic|KAMIS|425.96\*|--|37.58m|
> |Gurobi|Exact|425.95|0.00%|26.00m|
> |-|-|-|-|-|
> |RL+Sampling|LwD|422.22|0.88%|18.83m|
> |RL+Sampling|DIMES|423.28|0.63%|20.26m|
> |UL+Sampling|GlowNets|423.54|0.57%|23.22m|
> |SL+Sampling|DIFUSCO (T_s=100)|425.14|0.19%|53m41s|
> |SL+Sampling|T2T (T_s=50,T_g=30)|425.18|0.18%|38m1s|
> |SL+Sampling|OptCM (T_s=5,T_g=5)|**425.23**|**0.17%**|25m35s|
>
>
> 3. We supplement cross-dataset generalization results between RB graphs and ER graphs below. As seen, OptCM outperforms previous diffusion-based counterparts by a clear margin, e.g., in "Train:ER; Test:RB" "Sampling" setting, OptCM achieves significant performance gain from the previous (23.96%, 13m40s) to (10.64%, 2m37s).
>
> |Setting|Type|Method|Size|Drop|Time|
> |-|-|-|-|-|-|
> |Train:ER; Test:RB|Greedy|DIFUSCO (T_s=100)|15.87|21.00%|10m8s|
> |||T2T (T_s=50,T_g=30)|16.59|17.41%|15m5s|
> |||OptCM (T_s=1,T_g=1)|16.73|16.59%|40s|
> |||OptCM (T_s=5,T_g=5)|**17.01**|**15.21%**|2m39s|
> |Train:ER; Test:RB|Greedy|DIFUSCO (T_s=100)|29.98|27.54%|10m48s|
> |||T2T (T_s=50,T_g=30)|31.47|23.96%|13m40s|
> |||OptCM (T_s=1,T_g=1)|36.39|11.96%|43s|
> |||OptCM (T_s=5,T_g=5)|**36.94**|**10.64%**|2m37s|
>
> > **III. Is the CM used in this paper in the generalization or memorization regime?**
>
> Thank you for this very interesting question. OptCM indeed employs an atypical generative framework where we learn a high-quality solution distribution for a given instance, which can be viewed as a conditional generation process. However, since each condition (instance) is unique and the same instance conditions will not reappear during testing, the model cannot achieve plausible results by simply memorizing solutions for certain instance conditions. This is perhaps the intuitive response to your question. Nevertheless, we believe it is a rather open and intriguing question. Although we cannot provide empirical evidence due to the tight time window, we plan to investigate and analyze this question further in the future.

---

> ### Author Response · Authors · 2024-08-12
> **Official Comment by Authors (Cont.)**
>
> > **IV. How does the proposed method handle real-world graphs (such as SNAP graph in MIS)?**
>
> Thank you for the question. We would like to reiterate that this paper primarily focuses on TSP evaluation as the main experiment, with MIS results included as supplementary evidence to demonstrate the model's ability to handle different problems. This structure aligns with the organization of leading conference works like DIMES, DIFUSCO, and T2T, and it's worth noting that many studies focus solely on TSP, such as GNNGLS and UTSP. We believe that TSPLIB experiments effectively supports the real-world performance of OptCM on the TSP problem. As for the MIS problem, we add the experiments on SATLIB dataset and we believe the results effectively demonstrate the model's capility to generalize to real-world graphs.
>
> Regarding SNAP graphs, we faced challenges completing experiments within the tight time constraints. The majority of SNAP graphs contain millions of nodes, making it extremely time-consuming for heuristic solvers to provide ground truth within the allotted time. Additionally, our model was primarily trained on problems with fewer than 1,000 nodes, and scaling up to the size of SNAP graphs might exceed the current generalization capabilities of our learning-based model. We intuitively believe a large-scale training dataset (e.g., ER-[9000-11000]) is necessary to better generalize to SNAP graphs. Although we are unable to provide results at this time, we are happy to extend our work to address this dataset in the immediate future.
>
> ---
>
> Thank you for the advice and the opportunity to engage in this valuable discussion. We hope our response satisfactorily addresses your concerns and we will include all these meaningful empirical results and discussions in our final version.
>
> We believe this work advances the field by making important strides in improving NCO's performance, given OptCM's strong empirical results and the potential to serve as a more powerful backbone model for future works. We would sincerely appreciate it if you could reconsider your rating, and we are more than willing to address any further concerns you may have.

---

> > ### Comment · Reviewer_rPB6 · 2024-08-12
> > **Response to Authors - Final Remarks**
> >
> > Thank you for providing more results to evaluate your model and the additional clarification. I do acknowledge the following:
> >
> > - The use of CMs did improve the results in terms of solution quality and run-time when compared to other DM-based methods.
> >
> > - The sampling procedure inherited in pre-trained CMs (or generally DMs) at inference can be modified to be a per-instance solver and allow for conditional sampling. When compared to the inference of GNNs, I think the proposed approach has a better potential to further improve. I fully accept this point from the authors.
> >
> > - When compared to problem-specific heuristics, the method does achieve a significant improvement in terms of *inference* time due to the fast sampling of CMs and the use of GPUs. However, the method significantly under-performs when compared to these classic solvers in terms of solution quality.
> >
> > In short, under the setting that (1) a large accurately-labeled training set is accessible, (2) the massive "offline" training time can be ignored, and (3) the testing instances are not that different from the training set, the proposed solver can obtain "descent" solutions extremely fast.
> >
> > I am not suggesting that the proposed solver must outperform on every metric for every COP across all datasets. Instead, I am emphasizing that the authors should thoroughly investigate the limitations of their method and identify the setting in which it excels.
> >
> > A suggestion regarding SNAP graphs (very sparse and very large graphs): If the authors are still okay with labeling their training graphs using KaMIS (though I still have some reservations and believe this should be further investigated in subsequent versions of the paper), then KaMIS should be capable of obtaining solutions in a fair amount of time (albeit not guaranteed to be optimal), as it relies heavily on various graph reductions (see their results in the LwD paper for an example). Additionally, Google has introduced a SAT solver called CP-SAT (https://developers.google.com/optimization), which can solve very sparse/very large instances using the MIS ILP formulation significantly faster than Gurobi and KaMIS.
> >
> > Although I have doubts over the practicality of the settings where this method excels, which I believe would require further investigations, I will raise my score to 5 given the potential of further improvements on the per-instance conditional sampling in trained CMs.

---

> > > ### Author Response · Authors · 2024-08-13
> > >
> > > Thank you for acknowledging our work and raising the score, as well as for the insightful suggestions and valuable discussions. We fully concur that a thorough investigation of the limitations and a clear identification of the settings where our method excels are essential for the comprehensiveness of our study. We also appreciate the valuable suggestion for handling the SNAP dataset. We will incorporate all supplementary results and explanations into the final version of the paper and will continue to refine our work and add additional results based on your valuable feedback.

---

> ### Author Response · Authors · 2024-08-13
> **Kind Reminder**
>
> Thank you once again for your constructive feedback and your willingness to consider raising your score to 5. We greatly appreciate your support. We have noticed, however, that the updated score has not yet been reflected in the system. If it’s not too much trouble, could you kindly take a moment to update the rating at your earliest convenience? We greatly appreciate your attention to this detail, which is crucial for the final evaluation of our work. Thank you very much for your continued support and consideration.

---

### Official Review · Reviewer_P6jy · 2024-07-11

**Soundness:** 3
**Presentation:** 3
**Contribution:** 2
**Rating:** 5
**Confidence:** 4

**Summary:**

This paper presents Optimization Consistency Models (OptCM) for solving combinatorial optimization (CO) problems efficiently. By leveraging the consistency model, OptCM maps varying noise levels to optimal solutions in a single step, significantly reducing computational overhead. This approach is validated through extensive experiments on the Traveling Salesman Problem (TSP) and Maximal Independent Set (MIS), demonstrating significant superior efficiency compared with state-of-the-art diffusion-based models.

**Strengths:**

1. This paper introduces a consistency model to improve the efficiency of diffusion-based combinatorial optimization solvers.
2. Extensive experiments on TSP and MIS show that OptCM can outperform existing methods.

**Weaknesses:**

1. This work is mainly incremental, based on previous works DIFUSCO [1] and T2T [2].
2. Larger-size TSPs, such as those with 100000 nodes, should be tested against state-of-the-art learning-based methods [3].
3. Despite significantly improving solving efficiency, the proposed method is limited in addressing constrained COPs (e.g., CVRP and more complex COPs) and requires optimal solutions as labels.

[1] DIFUSCO: Graph-based Diffusion Solvers for Combinatorial Optimization, NeurIPS, 2023.

[2] T2T: From Distribution Learning in Training to Gradient Search in Testing for Combinatorial Optimization, NeurIPS, 2023.

[3] GLOP: Learning Global Partition and Local Construction for Solving Large-scale Routing Problems in Real-time, AAAI, 2024.

**Questions:**

See weaknesses.

**Limitations:**

The authors have discussed the limitations.

---

> ### Author Rebuttal · Authors · 2024-08-07
>
> Thanks for your valuable comment, nice suggestions, and for acknowledging our soundness, presentation, experimental extensiveness, and empirical performance. We seriously value the main novelty concern reflected in the comment and carefully address it in the general response. Below we respond to your specific comments.
>
> > **Q1: This work is mainly incremental, based on previous works DIFUSCO and T2T.**
>
> OptCM is inspired by some excellent previous works, which is one of the key supports for its ability to achieve such strong experimental results. However, we must note that the technical contributions of OptCM are highly non-trivial. Please kindly refer to the general response for the specific novelty claim, which we also briefly summarize here:
>
> 1. Compared to DIFUSCO and T2T's diffusion models, OptCM's model design, including the model inference process and the training objective function, is very different. Indeed, consistency models are recognized as a new independent class of generative models in generative tasks, while we further establish optimization consistency models (OptCM) to tailor the model for optimization scenarios.
> 2. The testing-phase gradient search method proposed by OptCM represents a completely novel approach compared to that of T2T. In fact, due to the difference between consistency mapping and diffusion inference, OptCM's framework and implementation are fundamentally independent.
> 3. In terms of experimental results, OptCM demonstrates exceptional performance, not only in solution efficiency but also in the quality of the solutions. We believe such strong empirical results substantiate the claim that OptCM represents more than just incremental progress within the existing diffusion framework.
>
> In addition, the reviews from reviewers 3pWn and rpwx firmly support our novelty claims: "Overall I think the method is novel. The extension of consistency training framework into diffusion-based CO solvers is not trivial, and this work is trying to solve a well-motivated problem. The empirical evaluations are quite convincing compared to diffusion-based CO solvers." “Overall, the paper's strengths lie in its innovative approach to reducing computational complexity while maintaining high solution quality, its robust and versatile model design, and its impressive performance on benchmark tasks.”
>
> > **Q2: Larger-size TSPs, such as those with 100000 nodes, should be tested against state-of-the-art learning-based methods like GLOP.**
>
> Thanks for the valuable question. In the rebuttal phase, we extend our model training to TSP-10000 and compare OptCM with several baselines, including DIFUSCO and GLOP.
>
> |Method|Length|Drop|Time|
> |-|-|-|-|
> |GLOP|75.62|5.36%|32s|
> |GLOP more revisions|75.29|4.90%|1.8m|
> |DIFUSCO(T_s=100)+2Opt|73.57|2.51%|31m24s|
> |OptCM(T_s=5)+2Opt|72.94|1.63%|15m35s|
>
> We also explore generalizing the model to 100k nodes to assess its scalability. However, we discover that although OptCM can achieve quick prediction in seconds, the greedy decoding procedure—which sequentially inserts edges or nodes with the highest confidence into the partial solution until a feasible solution is obtained—can be prohibitively time-consuming, making it meaningless for the evaluation. OptCM is not yet suitable for large-scale evaluations without an optimized decoding procedure or a high-level divide-and-conquer strategy like GLOP. Indeed we suggest that there may be a chance for the combination of OptCM's backbone model and GLOP's divide-and-conquer strategy for scalability.
>
> We will add the above evaluation to our paper and include the discussion of GLOP in related work and future work discussion to enhance the comprehensiveness of the paper.
>
> > **Q3: Limitations in addressing constrained COPs and supervision requirements.**
>
> In fact, our method can handle various constrained problems. We rely on the powerful expressive capability of generative models to learn the constraints, ensuring that the generated heatmap results are as close to feasible as possible. This is followed by post-processing (e.g., greedy algorithms) to utilize the model's output while strictly satisfying the constraints. This paradigm theoretically handles any constraint, which is also the approach used in previous mainstream works such as Diffusco and T2T. However, for more complex constraints, the model's ability to capture constraints may be limited. In such cases, using sequential models to explicitly control constraints at each step can be more effective. For example, the mainstream methods for the CVRP problem are still based on sequence models.
>
> Nevertheless, heatmap methods have their own advantages, such as not suffering from the sparse reward problem, enabling them to end-to-end handle larger-scale problems with better solution quality. These methods are currently the SOTA for classic problems like TSP and MIS, highlighting their strengths. There has also been considerable works in leading conferences that maintain competitive performance, demonstrating the value of this line of research. Additionally, currently works focusing on extending neural solvers for larger-scale problems using divide-and-conquer paradigms become a new important line of research.
>
> We believe that different methodological routes have their own expertise in dealing with problems of different characteristics, and each route has its own advantages and disadvantages. We will add these discussions to our paper, including a discussion of some recent works such as GLOP and LEHD [1].
>
> ---
> We hope this response can help address your concerns. We believe that this work contributes to this community and could potentially provide a more powerful backbone for future works in this domain. We would sincerely appreciate it if you could reconsider your rating, and we are more than happy to address any further concerns you may have.
>
> [1] Neural Combinatorial Optimization with Heavy Decoder: Toward Large Scale Generalization. NeurIPS 2023.

---

> > ### Comment · Reviewer_P6jy · 2024-08-10
> >
> > Thank you for your response. However, I still believe this method faces significant challenges when addressing constrained COPs, such as CVRP, even with its relatively simple constraints.
> >
> > 1. While these heatmap methods incorporate post-processing techniques to theoretically handle any constraints, their practical performance may be inadequate, as the model often struggles to fully capture these constraints.
> > 2. Due to the NP-hard nature of COPs with complex constraints, obtaining optimal solutions as labels remains impractical.
> > 3. Recent research [1] also questions the effectiveness of ML-based heatmap generation, suggesting that the performance is largely driven by the post-hoc search rather than the heatmap.
> >
> > [1] Position: Rethinking Post-Hoc Search-Based Neural Approaches for Solving Large-Scale Traveling Salesman Problems, ICML, 2024.

---

> ### Author Response · Authors · 2024-08-10
> **Thanks for the Prompt Reply**
>
> Thanks for the prompt reply. During the rebuttal process, we presented our novelty claim, supplemented large-scale results, and responded to the limitations of handling complex constraint problems. Despite this, the reviewer continues to express concerns regarding the limitations of heatmap-based methods, which we believe warrant meaningful discussion. Below, we offer our detailed response to this remaining concern.
>
> > **1: While these heatmap methods incorporate post-processing techniques to theoretically handle any constraints, their practical performance may be inadequate, as the model often struggles to fully capture these constraints.**
>
> We totally agree with this point. Relying on neural models to capture the constraints would require balancing the tradeoff between the model expressiveness and the task difficulty. When focusing on more complex constraints, sequence models will show their superiority in handling constraints by explicitly controlling constraints at each step. We will highlight this point in our final version.
>
> Nevertheless, we still need to note that heatmap methods have their own advantages, such as not suffering from the sparse reward problem, enabling them to end-to-end handle larger-scale problems with better solution quality. These methods are still the current SOTA for classic problems like TSP and MIS, highlighting their strengths. There have also been considerable heatmap-based works in leading conferences that maintain an impact on the community, like GCN, DIMES, DIFUSCO, T2T, etc., showing its value as an important research line in the Neural CO domain.
>
> Again, we note that different methodological routes have their own expertise in dealing with problems of different characteristics, and each route has its own advantages and disadvantages. We believe our method, with a novel model backbone and strong empirical performance, can contribute to this community and inspire future works to further enhance NCO.
>
> > **2. Due to the NP-hard nature of COPs with complex constraints, obtaining optimal solutions as labels remains impractical.**
>
> Since we use a generative model to learn the distribution of high-quality solutions, we do not impose strict requirements on the optimality of the training data labels. As demonstrated in Fig. 5 of T2T, training with suboptimal labels can still produce plausible solutions—surpassing the quality of the supervision—by sampling from the generative distribution. Indeed, in our experiments, we obtain the labels of large-scale problems from heuristic methods like LKH and KAMIS, which only require high solution quality rather than strict optimality to produce reasonable empirical results. We will include a discussion of supervision requirements and also mention this in the discussion of the comparison between RL and SL in our paper.
>
> > **3. Recent research [1] also questions the effectiveness of ML-based heatmap generation, suggesting that the performance is largely driven by the post-hoc search rather than the heatmap.**
>
> Please note that [1] primarily focuses on heatmap-based methods paired with strong post-hoc search techniques like MCTS. In contrast, our approach only employs basic decoding strategies, such as greedy decoding. Yet without MCTS, OptCM still demonstrates highly competitive performance, even surpassing LKH within limited time budgets (please see Figs. 3 and 4). In our settings, compared to previous heatmap-based methods like DIMES, DIFUSCO, and T2T, OptCM significantly outperforms them across all decoding settings, underscoring the substantial impact of heatmap quality on solving performance (please see Tables 1, 2, and 4).
>
> ---
> Thank you for the opportunity to engage in this valuable discussion. We hope our response effectively addresses your concerns. We believe this work advances the field by making important strides in improving NCO's performance. We would sincerely appreciate it if you could reconsider your rating, and we are more than willing to address any further concerns you may have.

---

> > ### Comment · Reviewer_P6jy · 2024-08-12
> >
> > Thank you for your response.
> >
> > I still believe that the contribution of this work is somewhat incremental, and the method may not be well-suited for constrained COPs. Nevertheless, I do see that the authors have made efforts to refine this work. Therefore, I am raising my score to 5. This neutral score means that if other reviewers and AC lean to accept this work, I would not oppose it. I encourage the authors to make more contributions in future version.

---

> ### Author Response · Authors · 2024-08-12
>
> Thank you for acknowledging our efforts and raising the rating. We will continue refining our work to make a meaningful contribution to the community. However, we would still like to emphasize our novelty and respond to the concern of the capability for other constrained COPs.
>
>
> **1) Novelty.**
>
> Indeed CMs are recognized as a new independent class of generative models in generative tasks, and in this paper, we further introduce the optimization consistency condition and establish optimization consistency models (OptCM) to tailor the model for optimization scenarios.
>
> Compared to diffusion solvers, the difference between the consistency mapping ($x_t\to x_0$) and the denoising function ($x_t\to x_{t-1}$) makes all the designs including model inference process, the training objective function, the gradient search procedure, etc., totally different. It is challenging to design such a new system with strong empirical performance.
>
> Moreover, the experimental results can support our claims. Our improvements over previous SOTA diffusion-based methods are of a significant magnitude, e.g., OptCM variants with more sampling and gradient search steps can achieve 82.1% performance gain with 14.7x speedup compared to previous diffusion-based counterparts on TSP-50/100. We believe that such strong empirical results substantiate the claim that OptCM represents more than just incremental progress within the existing diffusion framework.
>
> We affirm our commitment to making the source code publicly accessible upon publication. The objective of this work is to provide reference and contributions for the community. Open-sourcing the code is a fundamental obligation we undertake.
>
> **2) Concern of the capability for other constrained COPs.**
>
> As we previously mentioned, OptCM is theoretically capable of addressing any problem that can be framed as an edge-selecting or node-selecting problem. However, it is unrealistic to expect a single model to excel across all tasks. Different problems possess unique characteristics that may be better suited to specific methodologies. For problems like TSP, heatmap-based methods (using only basic greedy decoding) can achieve state-of-the-art performance, whereas RL-based approaches often struggle with sparse rewards and training instability, which may impede direct scaling to larger problem sizes. However, for problems with more complex constraints, sequence models demonstrate superiority by explicitly managing constraints at each step. Indeed, we do not yet know whether strong generative models can directly handle problems like CVPR; perhaps they already can, or at least have the potential to. (Though the discussion period is approaching its end, we will continue to adapt the framework to the CVRP task to validate this point.) The key point is that each methodological approach has its own strengths and weaknesses, and we do not expect the certain model to excel across all tasks. Indeed, there are also many papers in leading conferences merely focus on addressing a single problem, e.g., for TSP [1,2,3], MIS [4,5], SAT [6,7,8], etc., yet these contributions are still highly valued by the community.
>
>
> [1] H-tsp: Hierarchically solving the large-scale traveling salesman problem. AAAI, 2023.
>
> [2] Graph Neural Network Guided Local Search for the Traveling Salesperson Problem. ICLR, 2022.
>
> [3] Unsupervised Learning for Solving the Travelling Salesman Problem. NeurIPS, 2023.
>
> [4] Learning What to Defer for Maximum Independent Sets. ICML, 2020.
>
> [5] Maximum Independent Set: Self-Training through Dynamic Programming NeurlPS, 2023.
>
> [6] Learning a SAT solver from single-bit supervision. NeurIPS, 2019.
>
> [7] NSNet: A General Neural Probabilistic Framework for Satisfiability Problems NeurIPS, 2022.
>
> [8] Online Bayesian Moment Matching based SAT Solver Heuristics. ICML, 2020.
>
>
> ---
> Thank you again for the advice. We hope our response satisfactorily addresses your remained concerns. We believe OptCM advances the field by making important strides in improving NCO's performance, given OptCM's strong empirical results and the potential to serve as a more powerful backbone model for future works. We appreciate your previous acknowledgment and hope that our clarifications will merit a higher rating for our work.

---

### Official Review · Reviewer_3pWn · 2024-07-12

**Soundness:** 3
**Presentation:** 3
**Contribution:** 3
**Rating:** 6
**Confidence:** 4

**Summary:**

This paper introduced a new algorithm for solving some classic combinatorial optimization problems. The method falls into the category of learn-based generative solvers. More specifically, it is a direct extension of the DIFUSCO [1] and T2t [2] solver, which are diffusion-based generative solvers. The improvement is mostly done through improving on the sampling step of the two aforementioned works with consistency models (CM) [3], a recent notable regime that enables drastic reduction of the number of function evaluations (NFE, or sampling steps) of vanilla diffusion models. The novelty lies in extending CM into discrete regime and combining it with consistency-based gradient search, which is necessary for combinatorial optimization problem. Empirical evaluations show the effectiveness of this new solver, where it achieves competitive objective value in a much shorter time, compared to various baselines.


[1] Z. Sun and Y. Yang, “DIFUSCO: Graph-based diffusion solvers for combinatorial optimization, in Thirty-seventh Conference on Neural Information Processing Systems, 2023.

[2] Y. Li, J. Guo, R. Wang, and J. Yan, “T2t: From distribution learning in training to gradient search in testing for combinatorial optimization,” in Advances in Neural Information Processing Systems, 2023.

[3]  Y. Song, P. Dhariwal, M. Chen, and I. Sutskever, “Consistency models,” arXiv preprint arXiv:2303.01469, 2023

**Strengths:**

1. Overall I think the method is novel. The extension of consistency training framework into diffusion-based CO solvers is not trivial, and this work is trying to solve a well-motivated problem. The empirical evaluations are quite convincing compared to diffusion-based CO solvers (of course, one expects so as consistency models inference time is much faster than diffusion generative models).

2. The paper is overall well-written, and the authors seem to have done quite thorough literature reviews to gather sufficient baselines to compare to the performance of their work to.

**Weaknesses:**

1. It is necessary to elaborate on why the authors wrote "... note F1 is exactly the (implicit) the objective of the diffusion and consistency models..." at line 259. In other words we need to see why (should be a lemma with proof) we have the quantity $F_1$ in equation (6) is the equivalence of the loss in eq (4).

2. It is unclear how the overall training paradigm takes place in practice, including the gradient search part. The authors should include an algorithm box on the training of their OptCM framework. Algorithm 1 on Multistep Consistency Sampling is almost the same as in the original Consistency Models paper, so I suggest the authors move them to the Appendix. If I'm not mistaken, the training of consistency models is quite tricky, for example, it is crucial to design a suitable training time discretization schedule for CM to work well. Could the authors elaborate on this for their problem?

 I'm willing to re-evaluate my score if the authors can answer these two points.

**Questions:**

See weakness.

**Limitations:**

Consistency training introduced additional training overhead; I think the authors did not discuss this in Appendix D.

---

> ### Author Rebuttal · Authors · 2024-08-07
>
> Thank you for your valuable comments and insightful suggestions, as well as for acknowledging our novelty, motivation,  non-trivial contributions, and convincing evaluations. Your questions and suggestions are instrumental in further strengthening our paper. Below, we respond to your specific comments.
>
> > **Q1: Why we have the quantity F1 in equation (6) is the equivalence of the loss in eq (4)?**
>
> Thanks for your valuable suggestion. We will supplement the derivation and intuition to the paper.
>
> Recall that $F_1=-\mathbb{E}\_{q(\mathbf{x}|G)q(\mathbf{h}|\mathbf{x},G)}\left[\log p_\theta(\mathbf{x},\mathbf{h}|G)- \log q(\mathbf{x})q(\mathbf{h}|\mathbf{x},G)\right]$, we apply an approximation to the posterior over $\mathbf{x}=\mathbf{x}\_0$ as a point estimate $q(\mathbf{x}|G)=\delta(\mathbf{x}-\mathbf{\eta})$ and obtain $F_1=\mathbb{E}\_{q(\mathbf{h}|\eta,G)}\left[\log\frac{q(\mathbf{h}|\mathbf{\eta},G)}{p_\theta(\mathbf{\eta},\mathbf{h}|G)}\right]$. This term is exactly the usual variational bound on negative log likelihood conditioned on graph $G$, which is exactly the optimization objective of the diffusion models. It can be further derived under the denoising diffusion framework similarly in the original DDPM paper:
> $$\begin{aligned}F\_1&=\mathbb{E}\_{q(\mathbf{x}\_{1:T}|\eta,G)}\left[\log\frac{q(\mathbf{x}\_{1:T}|\mathbf{\eta},G)}{p\_\theta(\mathbf{\eta},\mathbf{x}\_{1:T}|G)}\right] \\\\ &= \mathbb{E}\_{q} \left[ \log \frac{\prod\_{t=1}^T q(\mathbf{x}\_t | \mathbf{x}\_{t-1},G)}{p\_\theta(\mathbf{x}\_T) p\_\theta(\eta | \mathbf{x}\_1, G) \prod\_{t=2}^T p\_\theta(\mathbf{x}\_{t-1} | \mathbf{x}\_t, G)} \right] \\\\ &= \mathbb{E}\_q \left[ -\log p\_\theta(\mathbf{x}\_T) + \sum\_{t=2}^T \log \frac{q(\mathbf{x}\_{t}|\mathbf{x}\_{t-1},G)}{p\_\theta(\mathbf{x}\_{t-1}|\mathbf{x}\_t, G)} + \log \frac{q(\mathbf{x}\_1|\eta, G)}{p\_\theta(\eta|\mathbf{x}\_1, G)} \right]            \\\\ &= \mathbb{E}\_q \left[ -\log p\_\theta(\mathbf{x}\_T) + \sum\_{t=2}^T \log \left( \frac{q(\mathbf{x}\_{t-1}|\mathbf{x}\_t, \eta,G) }{p\_\theta(\mathbf{x}\_{t-1}|\mathbf{x}\_t,G)}\cdot \frac{q(\mathbf{x}\_t|\eta,G)}{q(\mathbf{x}\_{t-1}|\eta,G)} \right) + \log \frac{q(\mathbf{x}\_1|\eta,G)}{p\_\theta(\eta|\mathbf{x}\_1,G)} \right] \\\\ &= \mathbb{E}\_{q} \left[ \log \frac{q(\mathbf{x}\_T|\eta,G)}{p\_\theta(\mathbf{x}\_T)} + \sum\_{t=2}^T \log \frac{q(\mathbf{x}\_{t-1}|\mathbf{x}\_t, \eta,G)}{p\_\theta(\mathbf{x}\_{t-1}|\mathbf{x}\_t,G)} - \log p\_\theta(\eta|\mathbf{x}\_1,G) \right] \\\\ &= \mathbb{E}\_{q} \left[ D\_{KL}(q(\mathbf{x}\_T|\eta,G) \| p\_\theta(\mathbf{x}\_T)) + \sum\_{t=2}^T D\_{KL}\left(q(\mathbf{x}\_{t-1}|\mathbf{x}\_t, \eta,G) \| p\_\theta(\mathbf{x}\_{t-1}|\mathbf{x}\_t,G)\right) - \log p\_\theta(\eta|\mathbf{x}\_1,G)\right] \\\\ &= \sum\_t \mathbb{E}\_{q} \left[D\_{KL}\left(q(\mathbf{x}\_{t-1}|\mathbf{x}\_t,\eta,G) \| p\_\theta(\mathbf{x}\_{t-1}|\mathbf{x}\_t,G)\right)\right] + C \end{aligned}$$
> This term is the optimization objective on point estimate $\eta$. While in optimization consistency models, we do not directly predict $p\_\theta(\mathbf{x}\_{t-1}|\mathbf{x}\_t,G)$ in the objective calculation, but we can virtually estimate this term using an additional predicted  $\hat{\eta}$ :
> $$\begin{align}p\_\theta(\mathbf{x}\_{t-1}|\mathbf{x}\_t,G)=q(\mathbf{x}\_{t-1}|\mathbf{x}\_t,\hat{\eta},G)=q(\mathbf{x}\_{t}|\mathbf{x}\_{t-1},\hat{\eta},G)\frac{q(\mathbf{x}\_{t-1}|\hat{\eta},G)}{q(\mathbf{x}\_t|\hat{\eta},G)}.\end{align}$$
> Then $F\_1$ can be reduced to the distance between the estimated $\hat{\eta}$ and the real $\eta$, i.e.,
> $$\begin{align} F\_1 = \sum\_t \mathbb{E}\_{q} \left[d\big(p\_\theta^t(\eta),\delta\left(\mathbf{x}-\mathbf{\eta}\right)\big)\right] + C,\end{align}$$
> which is equivalent to the consistency loss which optimizes points from different noise level to map to the target point.
>
> > **Q2: The authors should include an algorithm box on the training of their OptCM framework.**
>
> Thanks for the nice advice. We have included specific training algorithms in the PDF attachment of the general response and will incorporate them into our paper.
>
>
> > **Q3: More discussions on training details.**
>
> We supplement the specific design choices of our OptCM, and the listed hyperparameters correspond to those used in the algorithm presented in the paper. We will supplement these details to the paper to enhance its comprehensiveness.
>
> Moreover, we affirm our commitment to making the source code publicly accessible upon publication. The objective of this work is to provide reference and contributions for the community. Open-sourcing the code is a fundamental obligation we undertake.
>
> |Train|Design Choice|
> |-|-|
> |Consistency Loss Function|$d(x,y)=$Binary_Cross_Entropy$(x,y)$|
> |Scaling Factor|$\alpha=0.5$|
> |Weighting Function|$\lambda (t)=1$|
> |Discretization Curriculum|$t\sim \{1, 2,...,T\}$, randomly sampling|
> |Initial Learning Rate|$\eta=0.0002$|
> |Learning Rate Schedule|Cosine decay, decay rate $\omega=0.0001$|
>
> |Test|Design Choice|
> |-|-|
> |Sampling Step Schedule|$t_1=T(1-\sin(N\cdot i\pi/2))$, $t_2=T(1-\sin(N\cdot (i+1)\pi/2))$|
> |Guided Weighting Parameters|$\lambda_1=50$, $\lambda_2=50$ on TSP  <br> $\lambda_1=2$, $\lambda_2=2$ on MIS|
> |Rewrite Ratio| $\epsilon=0.2$ on TSP and ER-[700-800] <br> $\epsilon=0.3$ on RB-[200-300]|
>
> > **Q4: Consistency training introduced additional training overhead; I think the authors did not discuss this in Appendix D.**
>
> Thanks for your careful review and suggestion. Since the consistency model requires two inference predictions with different noise levels during training, it requires twice the training cost of the original diffusion model. However, this overhead on training is offline, and the consistency model is much more efficient than diffusion at inference time. We will supplement this in our new revision.
>
> ---
>
> We hope this response could help address your concerns, and we are more than happy to address any further concerns you may have.

---

> > ### Comment · Reviewer_3pWn · 2024-08-12
> >
> > Thank you for the extensive rebuttal. I am happy to keep my current score, which is leaning acceptance.

---

> > > ### Author Response · Authors · 2024-08-12
> > >
> > > Thank you very much for your thoughtful feedback throughout the review process. We are pleased to hear that you found our rebuttal satisfactory, and we are making every effort to improve the paper owing to your valuable feedback. We believe OptCM advances the field by making important strides in improving NCO's performance, given OptCM's strong empirical results and the potential to serve as a more powerful backbone model for future works.
> > >
> > > Given your earlier indication that a re-evaluation of the score was possible based on our response, do you have any other concerns keeping you from raising your score? If any other concerns remain, we are more than willing to offer clarifications or engage in further discussions. Your continued support would be greatly appreciated, as it would significantly bolster our efforts. Thank you once again for your time and consideration.

---

> > > > ### Author Response · Authors · 2024-08-14
> > > > **Gentle Reminder for Possible Re-evaluation**
> > > >
> > > > Dear Reviewer 3pWn,
> > > >
> > > > Thanks again for your time and effort involved in the review process. As the discussion period is approaching its conclusion, we want to respectfully remind you of our last correspondence. Given your earlier indication that a re-evaluation of the score was possible based on our response, if there are no further concerns, might we kindly ask you to consider reassessing your score? Your continued support would be greatly appreciated, as it would significantly bolster our efforts. Thank you once again for your time and consideration.

---

> > > > > ### Comment · Reviewer_3pWn · 2024-08-14
> > > > >
> > > > > Dear authors,
> > > > >
> > > > > I respectfully request you to stop diluting the review section of the submission by constantly posting comments pressing me to increase the evaluation.
> > > > >
> > > > > I think my previous comment has clearly stated that I will keep my current score as final, there is no need to send several messages. The main reason for this decision is I think the score reflects well the my view of the work, and as well as after taking into accounts all other reviews.
> > > > >
> > > > > Best regards,
> > > > >
> > > > > Reviewer 3pWn.

---

> > > > > > ### Author Response · Authors · 2024-08-14
> > > > > >
> > > > > > Dear Reviewer 3pWn,
> > > > > >
> > > > > > Thank you for your quick reply and for clarifying your position regarding the evaluation of our submission. We apologize if our previous communications appeared to overstep; it was certainly not our intention to dilute the review process but rather to ensure all concerns were thoroughly addressed.
> > > > > >
> > > > > > We respect your decision to maintain the current score and appreciate the time and consideration you have given to our work. Your insights have been invaluable to us, and we will take them into account as we continue to refine our work.
> > > > > >
> > > > > > Thank you once again for your contributions to the review process.
> > > > > >
> > > > > > Best Regards,
> > > > > >
> > > > > > The Authors

---

### Author Rebuttal · Authors · 2024-08-07

We sincerely appreciate the reviewers’ time, valuable feedback, and constructive suggestions. Overall, the reviewers deem our work as "well-motivated" (3pWn), "well-written" (3pWn, rPB6), and "technically sound" (rPB6), acknowledging our "novel" "robust" "versatile" methodology and model design (3pWn, rpwx), with "convincing" "significantly superior" "extensive" "impressive" empirical evaluations (3pWn, P6jy, rpwx). The primary concerns revolve around the novelty and inherent limitations in generalization associated with the data-driven learning paradigm, where fitting a specific data distribution may hinder generalization to different distributions. To preempt the potential misunderstandings that might impact the evaluation of our work, we first restate our contributions and address the generalization concerns.


**(1) Novelty Claim.**

OptCM is inspired by some excellent previous works, which is one of the key supports for its ability to achieve such strong experimental results. However, we must note that the technical contributions of OptCM are highly non-trivial, as supported by reviewers 3pWn and rpwx.

1. Compared to DIFUSCO and T2T's diffusion models, OptCM's model design, including the model inference process and the training objective function, is very different from the previous diffusion model. Indeed, consistency models are recognized as a new independent class of generative models in generative tasks, and in this paper, we further introduce the optimization consistency condition and establish optimization consistency models (OptCM) to tailor the model for optimization scenarios. This stands the model to be a new highly effective and efficient backbone for future development of learning-based solvers.
2. In terms of T2T's two-stage framework, we merely refer its high-level concept to introduce instance-tailored solving during testing, thus bridging the gap between training and problem-solving. In fact, due to the difference between consistency mapping and diffusion inference, OptCM's framework and implementation are fundamentally independent. The testing-phase search is an additional effort building upon the original inference model. With the consistency mapping-based model, the search procedure is an innovative method that is completely different from that of T2T.
3. For experimental results, OptCM demonstrates exceptional performance in both solution efficiency and quality. Notably, our improvements over previous SOTA diffusion-based methods are of a significant magnitude, e.g., OptCM variants with more sampling and gradient search steps can achieve 82.1% performance gain with 14.7x speedup compared to previous diffusion-based counterparts on TSP-50/100. Additionally, this is the first time end-to-end solvers have surpassed LKH on the runtime-gap curve for the TSP problem within a given timeframe. We believe that such strong empirical results substantiate the claim that OptCM represents more than just incremental progress within the existing diffusion framework.


**(2) Generalization Discussion.**

The major concern of Reviewer rPB6 is about the application of data-driven machine learning in problem solving, which might cause the model to fit a specific data distribution and limit its generalization to different distributions. Here we provide a targeted discussion.

Firstly we must note that Generalization challenges are common across data-driven machine learning methods, not just ours. Any machine learning method that automatically learns solving strategies and data structures from training data (whether supervised, reinforcement learning, or unsupervised) faces generalization challenges. Nonetheless, Machine Learning for Combinatorial Optimization (ML4CO) has gained significant traction, evidenced by numerous publications at leading conferences such as NeurIPS, ICML, and ICLR—each annually featuring about 20 papers on this subject. Please refer to [1] for a summarized list of ML4CO papers, including 34 CO problems and problems like TSP has more than 50 papers recorded. Around 20 survey papers, including Yoshua Bengio's first-authored paper [2], provide extensive summaries and highlight ML4CO's critical research value. Therefore, considering the demonstrated success in the ML4CO field, it would be inappropriate to evaluate a method primarily on its machine learning attributes.

On the other hand, learning from the given data is indeed one of the advantages of learning-based solvers. Data in real-world applications often exhibits a certain problem structure within the implicit distribution. In many cases, we need machine learning methods to uncover patterns within the data and solve problems associated with a specific distribution. It is also supported by Bengio in [2] that "ML4CO focuses on CO algorithms that automatically perform learning on a chosen implicit distribution of problems", and "ML can help improve an algorithm on a distribution of problem instances in two ways: 1) replace some heavy computations by a fast approximation; 2) explore the decision space and learn out of the experience the best-performing behavior (policy)."

Regarding the empirical results, we have already performed many experiments on the generalization results, including generalizing to different scales of TSP (from 50 to 1000)  as shown in Sec. 6.1 and Table 3, and generalizing to different distributions like real-world TSP instances from TSPLIB as shown in Appendix A and Table 1, 2. These results show that OptCM maintains good generalization results and outperforms previous SOTA diffusion-based baselines. We also supplement generalization results on MIS to strengthen our paper in the specific individual comments. On the other hand, generalization is itself an independently important research domain, which can typically be orthogonal to our method.

[1] github.com/Thinklab-SJTU/awesome-ml4co

[2] Machine Learning for Combinatorial Optimization: a Methodological Tour d'horizon. EJOR 2021.

---

### Decision · Program_Chairs · 2024-09-25

**Decision:**

Accept (poster)

**Comment:**

This was a borderline paper.
The authors introduce a new method for solving combinatorial optimization problems like TSP and Max Independent Set. Previous recent works (DIFUSCO and T2T) introduced diffusion solvers for combinatorial optimization and this paper introduces Consistency to accelerate the samplers for these discrete domains. This is a very reasonable thing to do, and some reviewers considered it somewhat incremental. Still, the scientific evaluation is detailed and the performance benefits are valuable.
Also, the reviewers raised several questions ranging from training details to additional comparisons and additional combinatorial optimization problems.
There was an extensive discussion and the authors provided numerous additional details in the rebuttals. The Reviewers were not fully excited, but overall the paper merits acceptance esp. given the additional clarifications that the authors provided.